

# Analysis of functional groups in atmospheric aerosols by infrared spectroscopy: functional group quantification in US measurement networks

Matteo Reggente[1], Ann M. Dillner[2], and Satoshi Takahama[1]

[1]ENAC/IIE Swiss Federal Institute of Technology Lausanne (EPFL), Lausanne, Switzerland
[2]University of California – Davis, Davis, California, USA

*Correspondence to:* S.Takahama (satoshi.takahama@epfl.ch)

**Abstract.**

Peak-fitting (PF) and partial least squares (PLS) regression have been independently developed for estimation of functional groups (FGs) from Fourier Transform Infrared (FTIR) spectra of ambient aerosol collected on Teflon filters. PF is a model that quantifies the functional group composition of the ambient samples by fitting individual Gaussian line shapes to the aerosol spectra. PLS is a data-driven, statistical model calibrated to laboratory standards of relevant compounds and then extrapolated to ambient spectra. In this work, we compare the FG quantification using the most widely used implementations of PF and PLS including their model parameters, and also perform a comparison when the underlying laboratory standards and spectral processing are harmonized. We evaluate the quantification of organic FGs (alcohol COH, carboxylic COOH, alkane CH, carbonyl CO) and ammonium, using external measurements (OC by thermal optical reflectance and ammonium by balance of sulfate and nitrate measured by ion chromatography). Moreover, we explore the sensitivity of FG, OC, OM/OC predictions to a series of possible input parameters available for each algorithm. We evaluate our predictions using 794 samples collected in the Interagency Monitoring of PROtected Visual Environments (IMPROVE) network (USA) in 2011 and 238 laboratory standards from Ruthenburg et al. (doi:10.1016/j.atmosenv.2013.12.034, 2014). Each model shows different biases. Overall, estimates of OC by FT-IR show high correlation with TOR OC. However, PLS applied to unprocessed (raw spectra) appears to underpredict oxygenated functional groups in rural samples, while other models appear to underestimate aliphatic CH bonds and OC in urban samples. It is possible to adjust model parameters (absorption coefficients for PF and number of latent variables for PLS) within limits consistent with calibration data to reduce these biases, but this analysis reveals that further progress in parameter selection is required. In addition, we find that the influence of scattering and anomalous transmittance of infrared in coarse particle samples can lead to predictions of OC by FTIR which are inconsistent with TOR OC. We also find through several means that most of the quantified carbonyl is likely associated with carboxylic groups rather than ketones or esters. In evaluating state-of-the-art methods for FG abundance by FTIR, we suggest directions for future research.



# 1 Introduction

Atmospheric aerosol, also called particulate matter (PM), is made up of organic compounds, inorganic salts, trace elements, black carbon, and water, among other substances. Accounting for its total mass in terms of its speciated composition is desirable for regulatory and epidemiological reasons, and this goal poses a substantial challenge for environmental analytical

measurement. Organic compounds in particular can comprise 20–80% of the atmospheric aerosols by mass (Lim and Turpin, 2002; Zhang et al., 2007), but the large number of molecule types present in these samples eludes exhaustive characterization. Typical methods for characterizing this organic fraction includes quantification of total carbon by evolved gas analysis and mass fragment analysis by mass spectrometry (e.g., Rogge et al., 1993; Chow et al., 1993; Hallquist et al., 2009; Laskin et al., 2012). Alternatively, reconstruction of organic aerosol mass by functional group (FG) abundance in these mixtures has been

demonstrated to provide high recovery (Maria et al., 2003). In addition, describing OM in terms of FGs have been useful for source apportionment as it captures particular emission characteristics (e.g., hydroxyl groups in marine sprays and biogenic SOA, ketonic carbonyl from burning) as well as atmospheric processes (e.g., carboxylic acid formation from photooxidation) (e.g., Decesari et al., 2007; Russell et al., 2011; Liu, 2014). Recent work has demonstrated the capacity of FG analysis to bridge the gap between molecular speciation and atomic composition obtained by chromatography and mass spectrometry

measurements, and chemically-explicit model simulations (Ruggeri and Takahama, 2016; Ruggeri et al., 2016). FGs can be characterized by nuclear magnetic resonance spectroscopy (NMR), Raman spectroscopy, gas chromatography with mass spectrometry (GC-MS), liquid chromatography, reaction or derivitization and spectrophotometry, and Fourier Transform infrared spectroscopy (FTIR) (Maria et al., 2003; Decesari et al., 2007; Dron et al., 2010; Kalafut-Pettibone and McGivern, 2013; Craig et al., 2015; Ranney and Ziemann, 2016).

In this work, we focus on the application mid-infrared (mid-IR) spectroscopy with FTIR for this purpose. Some of the earliest work in studying organic aerosol composition in Los Angeles smog and synthetic smog generated in the laboratory was studied using FTIR (Mader et al., 1952), and this tool is often used qualitatively for identifying organic aerosol constituents and monitoring changes induced under controlled laboratory conditions (e.g., Hung et al., 2005; Presto et al., 2005; Fu et al., 2013; Kidd et al., 2014; Chen et al., 2016; Yu et al., 2017). In a quantitative capacity, FTIR has the potential to expand the

suite of measurements available from a single Polytetrafluoroethylene (PTFE) filter — in addition to gravimetric mass, X-ray fluorescence, ion chromatography — as it is inexpensive, non-destructive, and requires no additional sample preparation. In addition, it is in principle possible to obtain the abundance of major inorganic components of PM in addition to organic FGs from the same FTIR spectrum as described below. The essential principle of the technique is to record chemically-specific absorption bands resulting from dipole moment transitions induced by interaction of molecular vibrations with mid-IR radiation

(Harris and Bertolucci, 1989). Quantitative analysis of spectra is based on the Beer-Lambert law which ascribes a linear relationship between the abundance of a substance and the mid-IR absorbance at wavenumbers corresponding to the vibrational modes of discriminating molecular bonds (Griffiths and Haseth, 2007). However, this task is confronted with several challenges. Condensed-phase spectra can have broad, overlapping absorption bands due not only to irreducible decay of excited vibrational states (lifetime broadening), but also to the slight variations in resonant frequencies of similar bonds interacting



with local environments (heterogeneous broadening) (Kelley, 2013). Absorption intensities of the same FG can additionally vary according to the neighboring substituents of each FG (Allen and Palen, 1989; Maria et al., 2003). These issues are particularly salient in environmental samples, which contain a large number of bonds of the same type in different configurations. Furthermore, given that atmospheric aerosols are complex mixtures containing thousands of different compound types (Hamilton et al., 2004; Kroll et al., 2011), we must necessarily base our strategies for characterization on what we can interpret from simpler laboratory mixtures.

Despite these challenges, past researchers have been able to build calibrations for mid-IR spectra to generate predictions for PM constituents that have compared favorably with other methods. For instance, researchers have quantified non-organic substances collected directly on PTFE or by other means: ammonium sulfate and bisulfate (Cunningham et al., 1974; Cunningham and Johnson, 1976; McClenny et al., 1985; Krost and McClenny, 1992, 1994; Pollard et al., 1990a; Allen et al., 1994; Tsai and Kuo, 2006; Reff et al., 2007), ammonium nitrate (Bogard et al., 1982; Pollard et al., 1990a), silica dust (e.g., Foster and Walker, 1984; Weakley et al., 2014; Wei et al., 2017), and carbon (Pollard et al., 1990b; Kirchner et al., 2000). Quantification of organic aerosols in terms of their constituent FGs was pioneered by Allen and co-workers (Allen and Palen, 1989; Paulson et al., 1990; Pickle et al., 1990; Mylonas et al., 1991; Palen et al., 1992, 1993; Holes et al., 1997), and further developed by other researchers (e.g., Blando et al., 1998; Maria et al., 2002; Sax et al., 2005; Reff et al., 2007; Cain et al., 2010; Coury and Dillner, 2008; Day et al., 2010; Russo et al., 2014; Faber et al., 2017). These organic FGs include aliphatic CH, alcohol COH, carbonyl C=O, carboxylic COH, organonitrate $CONO_2$, organosulfate $COSO_3$, and amine NH. The FG analysis approach to quantification of organic matter (OM) typically requires two steps: 1) estimating molar abundances of individual bond types from measured absorbances, and 2) relating bond abundances to FG and carbon content such that the OM mass can be obtained from their summation. On the second point, Takahama and Ruggeri (2017) proposed that organic molecular mixtures can be conceptualized as a collection of functionalized carbon atoms, from which atomic composition for mass estimation can be derived. While non-carbon atoms can be apportioned to each FG uniquely, carbon atom estimation from FG measurement is less straightforward. The carbon atoms are first separated into detectable and non-detectable fractions based on the FGss associated with each carbon and the FG calibrations available. While the combinatorial growth in polyfunctional carbon types possible from any set of FGs precludes estimation of specific carbon type abundances from FG abundance, the detectable carbon abundance can be statistically estimated from the FGs. Building on the findings of their work, we primarily restrict the scope of this manuscript to the first point in OM characterization from FGs (estimation of molar abundance from measured absorbance).

Methods for quantification of bonds from FTIR spectra fall into two broad categories (Alsberg et al., 1997). The first is a physically-based approach in which spectra are decomposed into their underlying peak representations, and FG abundance is estimated by relating absorption peaks with their molar absorption coefficients. Constraints for Lorentzian, Gaussian, Voigt, or fixed absorption profiles for individual bonds are combined to represent each spectrum with the aim of faithfully reconstructing the signal in regions where absorbances from bonds are assumed to be present. Gaussian peaks have been most commonly used for atmospheric aerosol analysis under the assumption that the FG absorption peaks are the sum of many peaks from individual compounds. Uncertainties in analysis can arise from the prescription of peak constraints and their combined fit in a high-dimensional mathematical space, and the selection of molar absorption coefficients to be applied to each type of bond.





While peak constraints and absorption coefficients are derived from laboratory standards, their values can vary according to the specific compounds selected for study. The second approach follows a more direct reliance on data whereby a statistical calibration model is constructed via multivariate analysis; partial least squares (PLS) regression is a common example from this category (Martens and Næs, 1991). In this method, a model comprising a set of latent variables is trained on laboratory

standards comprising individual compounds and mixtures of multiple substances; the final set of calibration model parameters (i.e., regression coefficients) is thought to embody some combination of absorption profiles, interferences, and absorption coefficients necessary to make accurate predictions for samples similar to the calibration set. This method requires fewer constraints imposed by the operator (except the assumption of linearity — or possibility for linearization — with PLS) than PF. As with the PF approach, uncertainties arise in extrapolating calibration models to atmospheric samples, but the lack of

physical constraints can lead to a broader range of predictions if model parameters (number of latent variables for PLS) are not judiciously selected (Takahama and Dillner, 2015). However, faithful reconstruction of the spectrum by model latent variables is not required as in PF; the target is to extract features only as necessary for accurate quantification. The accuracy of predictions is predicated on the combination of laboratory standards used to construct an approximate representation of this complex mixture space. While several variants of PLS with different sets of constraints exist (e.g., non-negativity, smoothness) (Rosipal

and Krämer, 2006), the most common version primarily imposes orthogonality on a set of latent variables and estimates the regression coefficients to maximize covariance with the response vector (FG abundance). Both PF and PLS has their own merits along criteria such as interpretability, ease of calibration sample preparation, and extensibility. For instance, new functional groups can be incorporated into peak-fitting from single-peak calibrations which are then fitted together with the existing peaks, whereas PLS requires recalibration together with laboratory standards that include potential interferences. Additionally,

when scattering interferences are present in the sample, spectral preprocessing can extend applicability of the Beer-Lambert law (Rinnan et al., 2009). The PTFE filters prevalent in atmospheric sampling is one such substrate with significant non-analyte contributions to the signal that must be separated either explicitly (by background correction) or implicitly (by PLS) for successful analysis. PTFE fibers exhibit a broad scattering contribution to the signal (McClenny et al., 1985) upon which absorption of analytes is superposed. Blank subtraction alone is often insufficient to eliminate the contribution of scattering to

the signal for quantitative calibration because of variability in the PTFE signal that can arise due to specific filter characteristics, different orientation within the FTIR beam, or deformation due to sample collection (i.e., application of vacuum). However, a blank subtraction step to remove peculiar signatures of the PTFE spectrum followed by an additional adjustment by linear or polynomial curve-fitting has been used successfully in the past (Gilardoni et al., 2007; Takahama et al., 2013b). Alternatively, a non-parametric model can be used to fit and remove the scattering signal without prior subtraction of a blank filter spectrum

(Kuzmiakova et al., 2016). This preprocessing is required for the current implementation of the PF algorithm, while the latent variable modeling approach of PLS has been shown to provide accurate quantification of substances with and without an additional baseline correction step (Ruthenburg et al., 2014; Takahama and Dillner, 2015; Dillner and Takahama, 2015a, b). Additional baseline correction approaches for this task are surveyed by Kuzmiakova et al., 2016.

Given the range of methods available for manipulating and calibrating FTIR spectra, the objective of this paper is to eval-

uate the robustness in estimated abundances of aliphatic CH, alcohol COH, carboxylic COH, and carbonyl CO FGs and the





associated OM in ambient PM collected onto PTFE filters. We evaluate two contrasting models currently used for FG quantification in atmospheric PM. One is the peak-fitting model of Takahama et al. (2013b) (including their published absorption coefficients), and the other is the PLS model built on raw spectra Ruthenburg et al. (2014). To further understand the role of spectral pre-processing (baseline correction), laboratory standards, and algorithms for quantification, we use the same lab-

oratory standards as Ruthenburg et al. (2014) to derive new absorption coefficients for PF, and build a PLS calibration on baseline corrected spectra used by PF. 794 ambient samples in the Interagency Monitoring of PRotected Visual Environments (IMPROVE) network (identical to Ruthenburg et al., 2014) is used for evaluation of predictions by each model. In addition, we also consider our capability to quantify ammonium as an analyte explicity rather than treating it as an interferant (its NH stretching band prominently overlaps with the organic FG bands considered for this work). Kamruzzaman et al. (2018) found

that amine functional groups in the IMPROVE network contribute on average 5–15% to OM mass, ubiquitously, and without strong spatial or seasonal variations. For this reason, these values are accounted for in the mass balance by fixing concentrations to those reported by their work. Organonitrate peaks were not visible in most IMPROVE sample spectra, and are therefore not included in this study. Organonitrate FG contributions with FTIR measurements are reported to be <1–10% (Day et al., 2010; Russell et al., 2011; Corrigan et al., 2013; Takahama et al., 2013a; Rollins et al., 2013); their overall contribution to atmospheric

OM during specific sub-diel periods (Rollins et al., 2012; Fry et al., 2013; Ayres et al., 2015; Ng et al., 2017) may be masked by hydrolysis and underestimation by filter measurements (Liu et al., 2012).

To place the ambient predictions of the calibration models in context, collocated measurements of thermal optical reflectance organic carbon (TOR OC) and ammonium estimated using sulfate and nitrate (assuming fully neutralized) from anion ion chromatography are used as independent reference values for comparison. Both of these collocated measurements serve as

an upper bound for the FTIR predictions. Carbon atoms bonded only to other carbon atoms and to FGs that are not reported here can lead to an underestimation of carbon by 10–20% (Maria et al., 2003; Takahama and Ruggeri, 2017). The reference ammonium concentration is calculated by assuming that sulfate and nitrate measured on the collocated nylon filters are fully neutralized by ammonium, which does not account for the potential nitrate volatilization artifact from PTFE filters, presence of acidic aerosols, and association of nitrate with cations typically associated with mineral dust. These topics will be revisited

in the paper as pertinent to evaluation of our calibration models. Findings which are robust with respect to the method are presented, and recommendations for future development are provided to reduce uncertainty in the estimation of FG abundances.

## 2   Methods

The basis for quantitative spectroscopy can be described by the Bouguer-Lambert-Beer law (Griffiths and Haseth, 2007), which describes the attenuation of light as it travels through a medium. While many conventions for its expression exist in different

disciplines, we adopt the following notation for our application:

$$x_{ij} = \sum_{k=1}^{K} \epsilon_{jk} n_{ik}^{(a)} \tag{1}$$





$x$ is the absorbance (negative of the decadic logarithm of transmittance) specified for sample $i$ and wavenumber $j$; due to the sum of contributions from substances $k$. $n^{(a)}$ is the areal or surface density ($\mathrm{mol\,cm^{-2}}$), used in relation with suspended solids and thin samples (Duyckaerts, 1959; Nordlund, 2011), which draws parallels with PM mass measurement by beta attenuation (Kulkarni et al., 2011). $\epsilon$ is the (decadic) molar absorption coefficient ($\mathrm{cm^2\,mol}$) which completes this relationship. The aim of a calibration model is to solve the inverse problem of obtaining abundance of constituent substances giving rise to the observed absorbance.

In the following sections, we first describe laboratory and ambient measurements used for calibration and prediction (Section 2.1), algorithms for preprocessing (Section 2.2), sample clustering (Section 2.3), and calibration (Sections 2.4 and 2.5). The calibration models — specified through their training data, preprocessing method, calibration algorithm, and model selection — evaluated in this work are summarized in Table 1. We note that reference concentrations and calibration results are reported in units of $\mathrm{\mu mol\,cm^{-2}}$ for FGs in accordance with equation 1 and $\mathrm{\mu g\,cm^{-2}}$ for their related mass-equivalent quantities.

## 2.1 Experimental data

We use 794 IMPROVE network $\mathrm{PM_{2.5}}$ samples and 238 laboratory standard samples reported by Ruthenburg et al. (2014), and focus on the quantification of four organic functional groups and one additional inorganic group which absorbs in the same region: alcohol COH (aCOH), carboxylic COH (COOH), alkane CH (aCH), total (carboxylic, ketonic, and aldehydic) carbonyl (tCO), and inorganic ammonium N-H (iNH). We report the evaluation of predictions for urban and rural sites separately. The urban sites consist of two collocated measurement stations in Phoenix, AZ, while the rural sites consist of five locations: Mesa Verde, CO; Olympic, WA; Proctor Maple Research Facility, VT; Sac and Fox, KS; St. Marks, FL; Trapper Creek, AK, spread throughout the country.

Ambient samples were collected every third day from midnight to midnight (local time) for 24 hours. FTIR spectra are obtained for PM collected on $25\,\mathrm{mm}$ PTFE filters (Teflon, Pall Gelman – $3.53\,\mathrm{cm^2}$ sample area) of the same type that are analyzed for gravimetric mass, elements and light absorption in the IMPROVE network. The nominal flow rate is $22.8\,\mathrm{L\,min^{-1}}$, which yields a volume of $32.8\,\mathrm{m^3}$ for 24 hours. A Tensor 27 Fourier transform infrared (FT-IR) spectrometer (Bruker Optics, Billerica, MA) equipped with a liquid-nitrogen-cooled wide-band mercury cadmium telluride detector is used to analyze the PTFE samples in transmission mode, using the empty sample compartment as the background. The sample compartment is continuously purged with air containing low levels of water vapor and $\mathrm{CO_2}$. Further details are provided by (Ruthenburg et al., 2014) and (Dillner and Takahama, 2015a). TOR OC (and EC) mass is measured on quartz filters collected in parallel to the PTFE samples using the IMPROVE_A protocol (Chow et al., 2007). The TOR OC values are also adjusted for positive artifacts due to organic vapor adsorption onto quartz fiber by subtracting the monthly mean blank values. Sulfate and nitrate concentrations are measured on nylon filters also collected in parallel and analyzed by ion chromatography. Elemental composition is measured by X-Ray Fluorescence (XRF). The atmospheric concentrations of OC, nitrate, sulfate, and elemental composition provided by these techniques (obtained from the Federal Land Manager Environmental Database, FED, http://views.cira.colostate.edu/fed/Default.aspx) are converted to equivalent areal mass densities on the PTFE filters using the filter collection area and actual sampled volume.





## 2.2 Spectral pretreatment

The baseline correction method of Kuzmiakova et al. (2016) is applied to both ambient and standard spectra for peak fitting
and PLS described below. This method uses smoothing splines (Reinsch, 1967) to model the baseline by regressing onto the
background regions (where no analyte absorption is expected), and interpolating through the analyte region. The calculated
baseline is subtracted from the high-frequency region ($>1500\,\mathrm{cm}^{-1}$) where stretching or bending modes of aCOH, aCH,
cCOH, carbonyl CO, and amine NH are present (Shurvell, 2006).A single parameter effectively controls the curvature of
the fitted baseline, and we select the value which minimizes the negative absorbance fraction (NAF). NAF represents the
contribution of negative analyte absorbance $\|\boldsymbol{a}_{A-}\|_1$ to the total analyte absorbance $\|\boldsymbol{a}_A\|_1$:

$$\mathrm{NAF} = \frac{\|\boldsymbol{a}_{A-}\|_1}{\|\boldsymbol{a}_A\|_1} \times 100\% \qquad (2)$$

where $\|\cdot\|_1$ denotes the 1-norm magnitude of a vector (summation of all absolute values of vector elements). NAF is calculated
across the entire wavenumber range in the analyte part of in a given segment, excluding the $CO_2$ absorbance band. Raw and
baseline corrected spectra are show in Figure 1.

## 2.3 Cluster analysis

Spectra similarity in baseline corrected ambient sample spectra are used to group samples into clusters, as originally presented
by Ruthenburg et al. (2014, appendix). We use Ward's hierarchical algorithm (Ward Jr., 1963), which has demonstrated useful
categorizations in previous studies with FTIR spectra (e.g., Russell et al., 2009; Takahama et al., 2011). Essentially, each
baseline corrected spectrum is normalized by its two-norm vector magnitude, and 20 clusters are selected to reduce the risk
of grouping dissimilar samples together. Seven samples were excluded prior to cluster analysis and manually placed in three
groups based on spectral similarity to known source profiles (Russell et al., 2011) [clusters 21 ($n=2$) and 22 ($n=1$)], or
because the relative contribution of noise in low concentration samples would interfere with the clustering procedure [cluster
23 ($n=4$)]. Clusters 19 ($n=21$) and 20 ($n=19$) were identified as being anomalous by Ruthenburg et al. (2014) when
comparing FTIR-estimated OC to TOR OC. In this work, we identify two additional clusters which contain samples with
anomalous predictions of organic FGs or iNH: clusters 7 ($n=26$) and 16 ($n=12$). Predictions and spectral characteristics for
these four clusters are discussed separately, while the rest of the samples, classified as "rest" ($n=706$), are used for general
25   evaluation.

## 2.4 Peak fitting

The method of peak fitting constructs a physically-based representation of absorbances based on equation 1, accounting for
lineshapes of spectral profiles resulting from absorption broadening. The fitted lineshapes are constrained to be non-negative
and within wavenumber limits derived from laboratory standards and ambient samples as described by Takahama et al. (2013b).
30   To represent the essence of the peak fitting algorithm in discretized notation, let $s$ denote a lineshape function defined over
wavenumbers $\tilde{\nu}$, and an arbitrary set of peak parameters $\theta$ for sample $i$ and bond $k$. The parameters are collectively estimated





by nonlinear least squares fitting of overlapping curves to $x$ to minimize the residual $e$ over specific regions of interest. The number of moles of bond $n$ is estimated as a product of the molar absorption coefficient and the integrated absorbance for each bond:

$$n_{ik}^{(a)} = \int_{\tilde{\nu}=-\infty}^{\infty} \epsilon(\tilde{\nu})s(\tilde{\nu},\theta_{ik})d\tilde{\nu} \approx \bar{\epsilon}_k \Delta\tilde{\nu} \sum_{j=1}^{M} q_j s(\tilde{\nu}_j, \theta_{ik}) \qquad (3)$$

$$x_{ij} = \sum_{k=1}^{K} s(\tilde{\nu}_j, \theta_{ik}) + e_{ij} \ . \qquad (4)$$

$q$ denotes quadrature coefficients for numerical integration. $\Delta\tilde{\nu} \equiv \Delta\tilde{\nu}_j$ for FTIR, so this term has been taken out of the summation. $\bar{\epsilon}$ corresponds to the integrated absorption coefficient (which we report in units of $\mathrm{cm^2\,\mu mol^{-1} \cdot cm^{-1} = cm\,\mu mol^{-1}}$), which better characterizes the intensity of a dipole transition than $\epsilon$ when the absorption band spans a range of wavenumbers (Atkins and de Paula, 2006). We note that the use of these units mark a departure from previous convention of incorporating the filter collection area into the effective absorption coefficients (Maria et al., 2003; Gilardoni et al., 2007; Takahama et al., 2013b) ($\pi/4 \cdot 1.0^2\ \mathrm{cm}^2$ in their work), but are preferred as they permit generalization across different filter sizes ($\pi/4 \cdot 2.12^2\ \mathrm{cm}^2$ for Ruthenburg et al., 2014). For Gaussian lineshapes, $\theta_{ik}$ may correspond to any number of relevant amplitude, location, and width parameters for each bond, and an analytical solution exists for its integral. For fixed absorbance profiles (e.g., cCOH), the peak parameter corresponds to a scaling coefficient.

Typically, calibration parameters are obtained from single-compound standards where attribution of absorption to individual functional groups are least ambiguous. Prediction in more complex mixtures are enabled by the concurrent fitting of multiple absorption peaks and invocation of mixing rules to arrive at a representative absorption coefficient. In this work, we retain the algorithm for apportioning the absorbance spectrum to various functional groups (Takahama et al., 2013b) and re-evaluate absorption coefficients using the calibration standards prepared by Ruthenburg et al. (2014). The apportionment protocol assumes that all FGs are present in each sample, which is a convenient approximation in atmospheric samples; for laboratory standards or specifc source samples, the FGs to be fitted is specified for each spectrum. Regressing integrated absorbance against areal density, we fit linear models without intercept to each compound or mixture of compounds in accordance with the Beer-Lambert law (the value of the slope is unaffected by the inclusion of an intercept in this data set). We retain coefficients only for regressions which the coefficient of determination ($R^2$) is greater than 0.9, and combine values from each compound or mixture $i$ into a single coefficient (to be applied to ambient samples) for each FG $k$ using the fractional number of samples as weights:

$$\epsilon_k = \sum_{i=1}^{N} w_i \epsilon_{ik}, \quad w_i = \frac{n_i}{\sum_{i=1}^{N} n_i}$$

These weights are selected to generate comparable models to PLS, and for this reason we also include estimates of absorption coefficients in multicomponent mixtures in addition to single-component standards when the fitting quality criterion is met. The estimates using the original absorption coefficients (Russell et al., 2009; Liu et al., 2009; Russell et al., 2010) will be referred to as *PFo*, and estimates using the recalibrated absorption coefficients will be referred to as *PFr*.



## 2.5 Multivariate calibration

Multivariate calibration is an alternative approach that is typically formulated as a linear regression problem, with the analyte concentration as the regressand (response variable) and absorbances used as regressors. In scalar notation, this relationship is written as:

$$n_{ik}^{(a)} = \sum_{j=1}^{M} x_{ij}\beta_{jk} + e_{ik} \tag{5}$$

$n^{(a)}$ is the areal density for sample $i$ and functional group $k$; $x$ is the spectral absorbance, $\beta$ is a wavenumber-specific regression coefficient, and $e$ is the residual term. The number of wavenumbers at which absorbances are available exceeds the number of samples available for calibration (several thousand versus a few hundred), and the autocorrelation in absorbances due to the broadening leads to an underdetermined, collinear problem. Therefore, equation 5 must be solved by techniques other than classical least squares regression. PLS regression (Wold et al., 1984) is a generalization of multivariate multilinear regression, and alleviates these problems by orthogonal projection and rank reduction (Geladi and Kowalski, 1986; Haaland and Thomas, 1988). Latent variables that maximize covariance with the response variable are found to model both the spectra matrix and response variables (FG abundances):

$$n_{ik}^{(a)} = \sum_{\ell=1}^{L} t_{i\ell}q_{k\ell} + f_{ik}$$
$$x_{ij} = \sum_{\ell=1}^{L} t_{i\ell}p_{j\ell} + g_{ij} \tag{6}$$

$\ell$ denotes the latent variable index, $p$ and $q$ are the loadings of $x$ and $n^a$, respectively, and $f$ and $g$ are their residuals. The linear regression coefficients $\beta$ in equation 5 are in turn derived from the loadings. In contrast to peak fitting, multi-component reference standards that span the space of chemical composition is desired for PLS so that the range of composition — of both analytes and interferents — anticipated in prediction samples are available to train the model (Massart et al., 1988). As it is not possible to fully reproduce the high-dimensional chemical space of ambient samples, the amalgam of aerosol mixtures prepared in the laboratory target the main features in this space.

A series of candidate models which satisfy equations 5 and 6 is obtained by varying $L$, the maximum number of latent variables (LVs) or factors. The Nonlinear Iterative Partial Least Squares (NIPALS) algorithm (Wold et al., 1983) is used to generate each model, and 10-fold Venetian blinds cross validation (Hastie et al., 2009; Arlot and Celisse, 2010) on the calibration set is applied to estimate corresponding root mean square of cross validation (RMSECV) values for the models. The minimum RMSECV solution is typically selected as the preferred model (defined by the value of $L$), but Takahama and Dillner (2015) found that this approach leads to overfitting with unrealistic results (i.e., extremely negative values) when extended to prediction of FGs in ambient samples. We therefore use the randomization test approach proposed by van der Voet (1994) to select the number of LVs. This method compares squared prediction errors across models with fewer LVs than the reference (minimum RMSECV) model and selects one for which the prediction is not significantly worse. This procedure




is applied to each functional group separately such that each model is independent of one another (referred to as "PLS1" in chemometrics nomenclature). The number of LVs selected for PLSr and PLSbc are presented in Table S1.

Though the interpretation of PLS models are less straightforward than peak fitting, it is possible to examine how models are weighting spectral variables (wavenumbers) and calibration samples for making predictions. The regression coefficients are difficult to interpret directly as their magnitudes must be interpreted in combination with absorbances. In addition, the value of the regression coefficients can also be either positive or negative; the latter are associated with interfering species (Haaland and Thomas, 1988) or oscillations that increase with the number of LVs used (Gowen et al., 2011). Therefore, different approaches are used for estimating the contribution of specific wavenumbers to predictions. Takahama et al. (2016) used sparse calibration approaches that eliminated uninformative wavenumbers and used importance weighting to identify absorption bands used by PLS models. Wavenumbers highlighted by FG calibration models were associated with absorption bands of the target FGs, while retaining similar prediction capability to the full wavenumber models presented here. The contribution of LVs to the explained variation and sum-of-squares of the spectra matrix and response variable are discussed in Section S4. The relationship between equation 5 and equation 1 is illustrated in Section S3.

## 2.6 OC estimation from FG abundance

The constituent molar abundance $n_a$ for atom $a$ is calculated from the moles $n_k$ of FG $k$ through a coefficient $\lambda_{ak}$ such that $n_a = \lambda_{ak} n_k$. From $n_a$ estimated for {C, O, H} in this work, the OC mass, OM/OC mass ratio, and O/C atomic ratios are obtained. While assignment of $\lambda_{ak}$ for non-carbon atoms is unambiguous, $\lambda_{C \cdot k}$ is depends on the assumed bonding configuration of polyfunctional carbon atoms (Takahama and Ruggeri, 2017). For instance, methylene (-CH$_2$-) carbon have a value of $\lambda_{C \cdot aCH} = 0.5$ (one mole of carbon for two aCH groups), while methyl carbon (-CH$_3$) have a value of $\lambda_{C \cdot aCH} = 0.33$. It is also possible for the same carbon atoms to be associated with both aCH and aCOH, or other FGs, which makes the selection of $\lambda$ less intuitive with increasing number of combinations; several statistical approaches are available for estimation in these instances. The only difference between Russell and coworkers (Russell, 2003; Takahama et al., 2013b) and Ruthenburg et al. (2014) is the value of $\lambda_{C \cdot aCOH}$; the former authors define the value as 0.5 and the latter authors define it as 0. Overall, the choice of this value this makes a ∼10% difference in the carbon estimate for this data set. For this work we adopt the value of 0.5 which is also supported by a recent analysis of modeled $\alpha$-pinene secondary organic aerosol (Ruggeri and Takahama, 2016; Takahama and Ruggeri, 2017) according to the Master Chemical Mechanism (Jenkin et al., 1997; Saunders et al., 2003) The full set of coefficients is provided in Appendix S1. We refer to the OC reconstructed through FG predictions as *FG OC* in this paper.

## 2.7 Quantification of carboxylic acid and non-acid carbonyl groups

While the carboxylic group comprises two molecular bonds, the abundance of carboxylic hydroxyl and carbonyl bonds are conventionally quantified separately with calibration models developed for their respective absorption bands. The carbonyl quantified in this way can include contributions from ketonic and aldehydic carbonyl because of their proximity in absorption bands that are difficult to resolve in environmental samples; the carboxylic hydroxyl cCOH and total carbonyl tCO are re-



apportioned to estimate abundance of carboxylic COOH groups along with non-acid (ketonic, aldehyde, and ester) carbonyl CO (written as naCO). Stoichiometrically, $n_{\text{COOH}}$ and $n_{\text{cCOH}}$ are equivalent, while $n_{\text{naCO}}$ is simply the difference between $n_{\text{tCO}}$ and $n_{\text{cCOH}}$ (equation S1). In principle, the exact molar composition in a mixture should meet the condition that the tCO is in excess of the cCOH ($n_{\text{tCO}} \geq n_{\text{cCOH}}$), with naCO content indicated by the tCO in excess of cCOH. To account

for random errors, Takahama et al. (2013b) recommend the averaging of cCOH and tCO to estimate COOH when $n_{\text{tCO}} \sim n_{\text{cCOH}}$. The estimated cCOH can be greater than tCO $n_{\text{cCOH}} < n_{\text{tCO}}$ if the integrated absorption area or absorption coefficient is misspecified for either FG. In the absence of additional information, Takahama et al. (2013b) assumes that the tCO is unmeasured due to shift in absorption frequency below that used in peak fitting, and base the COOH estimate on the cCOH. In such cases, an overall unmeasured fraction was assigned on a per-campaign basis to align tCO to cCOH abundances in prevous

studies, but in this work we apply this reasoning on a per-sample basis (i.e., $n_{\text{COOH}} \equiv n_{\text{cCOH}}$ and $n_{\text{naCO}} = \max\{0, n_{\text{tCO}} - n_{\text{cCOH}}\}$.

One strategy to avoid the apportionment of tCO to COOH and naCO is to build an alternative PLS regression model to predict naCO directly, rather than tCO. The known concentrations in laboratory standards are transformed according to equation S1 and provided as the response vectors to equation 5. In this work, the contributions to naCO in calibration standards are provided by

12-tricosanone and arachidyl dodecanoate, and therefore corresponds to ketone and ester CO (i.e., no aldehyde CO). Previous studies atomizing dissolved aldehydic compounds found that they were transformed into alcohols by aldol condensation, which is also a possible but not necessary outcome in the atmosphere (Takahama et al., 2013b) and depends on the presence of water and hydration constant of the compound. Nonetheless, we will refer to our new estimates as naCO under the assumption that aldehyde CO, if present in ambient samples, has a similar spectroscopic response to ketone and ester CO to the extent that we

can quantify them. The COOH calibration remains identical to that for cCOH since $n_{\text{COOH}} \equiv n_{\text{cCOH}}$, and the stoichiometric consistency for estimating OC, OM, OM/OC, and O/C ratios using different estimates of carbonyl are summarized in Appendix S1.

## 2.8   Evaluation metrics

Metrics such as mean error, mean bias, RMSE, and many others exist for intercomparison among measured and estimated

values. In this work, to quantify overall bias we use total least squares slope (obtained via major axis regression) which accounts for uncertainties in both quantities being compared (Ripley and Thompson, 1987), and the Pearson's correlation coefficient ($r$) to quantify the strength of linear relationship between the two quantities. All evaluation metrics provided are for samples labeled as cluster "rest," and excludes clusters 7, 16, 19, and 20.

## 3   Results and Discussion

We first report on differences among estimated absorption coefficients (Section 3.1). In Sections 3.2 and 3.3, we discuss evaluations of estimated quantities of using these absorption coefficients.



### 3.1 Estimation of absorption coefficients

Calibration curves and predicted concentrations according to the peak-fitting strategy outlined in Section 2 are shown in Figure 2. Regression parameters are included in Table 2. Absorption coefficients estimated for cCOH by two mixtures (italic values in Table 2): 1) 1-docosanol and suberic Acid, and 2) 1-docosanol, suberic acid, and adipic acid are not included in the calibration

because of their low $R^2$ values. Malonic acid samples are additionally excluded in the estimation of aCH as its concentration range is far below the rest of the laboratory standards (below typical loadings of atmospheric samples) and the slope is twice greater than the next highest value (italic values in Table 2). This difference biases the weighted absorption coefficient in a way that does not reflect the weighting of the PLS regression (including this value makes a 19.9% difference in the absorption coefficient). Overall, we used 97% of the laboratory standard samples, of which two-thirds were reserved for the calibration

(Figures S1 – S4 in Section S5). We can see that when the absorption coefficient for each respective category is applied to corresponding test set samples not used in the calibration (the remaining one-third of samples), predictions are within 6% of the reference values (Figure 2, bottom row).

In Figure 3, we compare the new absorption coefficients with those summarized by Takahama et al. (2013b) (specific citations described in Figure caption) adjusted for filter collection area. aCOH for 1-docosanol is the only FG and organic

compound for which we have a direct comparison; the value estimated for the absorption coefficient in this work is 3.6 times greater. This is due to different baseline correction methods and fitting procedures used by Gilardoni et al. (2007). When the same spectra preparation (smoothing spline baseline correction) is applied, the difference is two times greater but occurs in the same proportion for aCH absorption — i.e., both aCOH and aCH absorption coefficients for 1-docosanol spectra ($n = 3$) acquired by Ruthenburg et al. (2014) are twice that for spectra acquired by Gilardoni et al. (2007) ($n = 6$) when processed in

the same way, but the ratio of aCOH to aCH absorbances for each method are within 4%. This bias may partially due to the fact that the slope of the calibration curve is determined by a single influential point for the few pure 1-docosanol samples collected by Ruthenburg et al. (2014). However, the consistency of the single-point estimate with aCOH coefficients for other mixtures containing 1-docosanol (Figure 3) may suggest other differences that need to be investigated. For instance, the refractive index of the substrate may also affect the apparent absorbance in the limit of thin films (Hasegawa, 2017); similarity in optical

properties of the filter type may have to be considered in future studies. Single measurements of fructose, glucose, as well as nine other sugars, were combined to derive an overall absorption coefficient for saccharides by (Takahama et al., 2013b) (point estimates for fructose are effectively the same as the combined estimate, and glucose is 70% smaller; Russell et al., 2010, Table S2). We observe that the absorption coefficients for individual compounds in this work are higher than the previous collective estimate. While there was previously no calibration performed for ammonium sulfate for its quantification by PF, the single

ammonium sulfate sample used for removal of ammonium interference in the fitting procedure (introduced by Russell et al., 2009) carried a mass of 6.0 µg over a $a = \pi/4 \times 1.0^2$ cm$^2$ collection area (Takahama et al., 2013b), so this value is used to calculate a point estimate for its absorption coefficient. The recalibrated absorption coefficient is greater by a factor of 1.7. This is likely due to the combination of using a single-point value for estimating the coefficient.



Absorption coefficients for aCH vary by a factor of 3.2 (between 1.0 and 3.2, over ten compounds); for aCOH by a factor of 1.9 (between 19.8 and 37.7, over seven compounds); for cCOH by a factor of 1.6 (between 32.8 and 51.6, over three compounds), and for tCO by a factor 1.6 (between 10.0 and 16.1, over seven standards (Figure S11). Without informed strategies for parameter selection, the range of valid possibilities for these absorption coefficients imparts uncertainty in FG calibration.

Shown on the right side of the Figure 3 are averaged values reported by Russell and coworkers (Gilardoni et al., 2007; Russell et al., 2009, 2010; Takahama et al., 2013b) and this work. Both sets will be compared in the following sections. The FG absorption coefficient for aCOH and aCH estimated in this work are higher by a factor of 1.8 and 1.3 respectively, which leads to lower estimates for FG abundances. In contrast, the FG absorption coefficient for cCOH is lower than that reported by Russell et al. (2009) (factor of 0.8) and comparable to the one reported by Takahama et al. (2013b). The FG absorption coefficient for tCO is comparable with the one of Russell et al. (2009) and 1.4 times greater than the one reported by Takahama et al. (2013b).

### 3.2 Comparison of estimated OC and ammonium to external measurements

We first compare quantities for which we have an independent estimate (TOR OC and ammonium) to place our predictions in context. Individual contributions of FGs used to estimate OC are discussed in Section 3.3. Comparisons are stated for regular samples not belonging to anomalous clusters (Section 2) unless otherwise noted. Evaluation for anomalous clusters are discussed separately in Section 3.6. Figure 4 summarizes the comparison of predicted concentrations of OC using different sets of absorption coefficients (for PF), or different spectra pretreatment (for PLS). In general, the correlation between TOR OC and FG OC is high (r = 0.84–0.97) and typically greater for urban sites than rural ones. In the urban samples, OC estimated by PLSr is closer to TOR OC with an underprediction of 12%, while the other methods underpredict TOR OC by 34–50%. In the rural sites, the agreement with TOR OC is more varied with three models underpredicting TOR OC by 0–22% and PLSbc by 40%. In general, the consistent underprediction is expected on account of the undetectable carbon atoms by FTIR due to lack of functionalization or association solely with bonds we do not measure.

The difference between PFo and PFr is due to the systematic increase in absorption coefficients used by PFr compared with PFo, which decreases the molar abundance of FGs and consequently, the FG OC. This difference is particularly articulated by the absorption coefficient for aCH (1.73 against 1.31) as its mole fraction is over 60% regardless of estimation method used.

The differences between PLSr and PLSbc are more difficult to understand, but some interpretations can be made. First, systematic differences can occur in the way that laboratory standard and ambient sample spectra are baseline corrected as the absorbance regions are different. Also, baseline correction used in the PLSbc does not include frequency lower than $1500\,\mathrm{cm^{-1}}$, thus excluding the alkane peak around $1450\,\mathrm{cm^{-1}}$, which is likely being used for aCH estimation by the PLSr model (Takahama et al., 2016). PLSr may be erroneously incorporating some information regarding the scattering by supermicron particles in its prediction (Section 3.6). Notably, the samples labeled as anomalous by Ruthenburg et al. (2014) (clusters 19 and 20) are predicted more consistently in relation to TOR OC with PLSbc than PLSr, suggesting the baseline correction has partially removed spectral features from the raw spectra that lead to unexpected deviations in predictions. For the remaining samples, it is generally possible to find models using raw and baseline corrected spectra that produce similar predictions, and also those in




which FG OC agree better TOR OC, if different number of LVs are selected. While we do not explore all possible combinations of parameters exhaustively in this paper, an example of how comparisons of FG OC predictions vary with number of LVs of aCH for PLS with baseline corrected is given in Figure S16. The solution that best matches TOR OC (referred to as PLSbc*) from this evaluation is shown in Figure 5. The solutions in the neighborhood (±3 LVs) of the PLSbc models for each FG are

highly correlated, but vary in their slope by a factor of approximately 1.5 (Section S8). PLSbc* varies from PLSbc for aCH by only 2 LVs. It is possible that justification for use of alternate model parameters that improve agreement with TOR OC can be made, but given the large number of possibilities we restrict our evaluation of results primarily to those obtained by the protocols described in Section 2.

Figure 6 summarizes the comparison of ammonium concentrations predicted by FTIR with the value estimated as the cation

counterpart of sulfate and nitrate. The correlation in comparisons is strong for rural sites (r > 0.89) and moderately strong (r = 0.47–0.71) in urban sites for all models. Part of this difference may be that the dynamic range of ammonium in rural sites is twice the value of urban sites. While our estimated reference values are thought to be upper bound on account of our assumptions — no evaporation loss of ammonium nitrate from PTFE or nitrate association with dust instead of ammonium are not considered — the reference ammonium is overpredicted by the PSLr model at urban sites and PFo for both urban and rural

sites. The PFo overpredictions can be explained by the uncertainty on the absorption coefficient estimated using a single value. The overprediction in urban sites only by PLSr is less simple to interpret. One possibility is the scattering contribution to the FTIR spectra by large particles may be more significant for these samples, and this effect is reduced by baseline correction.

While ammonium quantification by FTIR has been the focus of past researchers (Johnson et al., 1981; McClenny et al., 1985; Allen et al., 1994; Krost and McClenny, 1994; Reff et al., 2007), the recent work of Russell, Dillner, and co-workers

focused on organic FG quantification, and an extensive evaluation for ammonium has not been performed. However, as the absorption bands of iNH overlap with aCOH, cCOH, and aCH, it is useful to know whether the fitted ammonium scales with an external measurement. Such a simultaneous evaluation of bonds is important for PF, since the IR absorption in each spectrum is apportioned to contributions from various bonds; overapportionment for one bond can lead to underapportionment for another. Based on the assessment with PFr using a better-characterized absorption coefficient, comparisons with the reference

ammonium values suggests that no gross overestimation or underestimation of the fitting is unlikely. Calibration models for PLSr and PLSbc used in this work are developed independently of one another (the "PLS1" approach); therefore, the predictive capability of one species is not strongly tied to another as with PF. In summary, the LVs in the ammonium calibration model are not necessarily the same as the organic FGs; the over- or underprediction of ammonium is less consequential to how we interpret the FG quantifications. An alternative, multimodel formulation ("PLS2") can provide estimates of both analyte and

interferents using the same set of LVs. The current decision to use PLS1 is based on the knowledge that PLS1 typically outperforms PLS2 as it is optimized for each target analyte (Martens and Næs, 1991), but in extrapolation (as in our use case) the physically consistency offered by PLS2 may confer benefits not conventionally recognized with such models.





### 3.3 Variations in estimated FGs

Figure 7 summarizes pairwise correlation coefficients and regression slopes (excluding anomalous clusters 7, 16, 19, and 20) of
FG abundances estimated using the methods discussed in Section 3.2. Individual scatter plots can be found in the supplementary
material (Figures S5–S9). Overall, aCH, iNH, and tCO, agree with fairly high correlation $r > 0.75$, but the agreement of aCOH

and cCOH — two FGs with broad absorption regions on account of the OH stretch — vary more significantly. The strong
agreement of iNH predicted by PLS and PF are particularly notable. While the calibration models for PLSr and PLSbc are
formulated to predict ammonium iNH, they are technically equivalent to a calibration model for ammonium sulfate as this is
the only substance with this bond in the calibration set. However, the comparison of predictions against PF which only uses
the NH stretching peak — which are spectroscopically similar between ammonium nitrate and sulfate — suggests that the PLS

models are likely using features that are specific to this absorption band that is common to both ammonium salts.

     First, we focus on the comparison between PFo and PLSr. In urban samples, aCH presents the highest correlation ($r =$
0.96) of all FGs, presumably, because the aCH absorption is unambiguous on account of the high abundance of hydrocarbon
compounds in urban areas. However, the slope (0.68) reveals a systematic difference between the two. For the rest of the organic
FGs in both urban and rural sites, PFo estimates are higher than PLSr; the regression slopes vary between 1.39 (aCOH in rural

samples) and 2.57 (cCOH in rural samples). Correlation coefficients in the PFr-PLSr comparison for any organic FG is similar
to the PFo-PLSr comparison; the only notable difference is the larger regression slope for cCOH (1.94 and 3.25 for urban and
rural samples against 1.53 and 2.57 respectively), due to the lower absorption coefficient applied to PFr than PFo (Table 2). The
cCOH and tCO estimated by PFr are still higher than PLSr (by 1.78 to 3.25). However, the underprediction of FG OC relative
to TOR OC (Section 3.2) can be explained by the lower concentrations of aCH and aCOH estimated with the recalibrated

absorption coefficients as they comprise more than 70% of the organic aerosol mass according to PF analysis (Section 3.5).
PFo and PFr predictions agree closely with PLSbc, likely because they use the same portion of the spectra. The organic FG
correlations varies between 0.7 (aCOH – rural samples) and 0.99 (tCO – rural sample). PFr predictions are closer to PLSbc
than PFo, with the exception of cCOH, since they use the same laboratory standard compounds. The correlation for iNH are
greater than 0.97 with slope close to one in the case of PFr and greater than 1.8 in the case of PFo, which indicate a systematic

bias due to the different absorption coefficient used (14.84 and 8.89 for PFr and PFo respectively).  The correlation in tCO
between PF and PLS are high ($r > 0.81$), potentially because of the narrow absorption band of the carbonyl FG. PF estimates
of tCO for urban samples increase in correlation with PLS estimates from 0.82 to 0.96 (with the slope approaching unity) when
baseline corrected spectra are used for PLS, suggesting that signal contributions (presumably from PTFE) interfering from the
quantification are effectively removed.

Within the broader scope of assessing uncertainty for each FG, we can consider that the estimated slopes can vary according
to the selection of absorption coefficient value for PF and the number of LVs for PLS. The range of absorption coefficients are
given in Section 3.1, with aCH having the largest range. If we examine the set of PLSbc solutions (varying only in number
of LVs) with a correlation coefficient greater than 0.95 with the selected solutions presented here, we find that the largest
variability is also in the aCH (ranges are shown in Figures S11 and differences among models in S12). Models with different





number of LVs reflect different weighting of calibration samples and their responses, so it is not surprising to find that the magnitude of uncertainties (reported in Section 3.1) are similar between PLS and PF for the same set of calibration standards.

### 3.4 Quantification of carboxylic acid and non-acid carbonyl

Figure 8 compares the number of areal density of tCO and cCOH predicted by the same methods. Both PLSr and PLSbc predict
similar abundances of tCO as cCOH, suggesting that most of the carbonyl is associated with carboxylic acid groups, and the non-acid fraction is small to negligible. While it is possible for both models to use the carbonyl absorption band, Takahama et al. (2016) suggests that the wavenumbers are weighted differently by the two models. There is noticeably more scatter in the relationship between tCO and cCOH from the PF predictions, with the presence of naCO difficult to identify in these samples. Furthermore, tCO abundance is systematically lower than cCOH for many samples (discernable beyond the extent of scatter).
Takahama et al. (2013b) hypothesized that this discrepancy could be due to underestimation of carbonyl in samples where the absorption band is shifted to lower frequencies. However, given that 1) the constraint $n_{\text{tCO}} \geq n_{\text{cCOH}}$ is met for the PLS estimates, and 2) the relative overprediction by PF in comparison to PLS is greater for cCOH than tCO in these samples, it is likely that it is cCOH that is overestimated by PF for these samples. This may be due to the peak profile for cCOH, or baseline correction artifacts. For clusters 19 and 20 in which the overestimation is more severe, the baseline correction artifact is the
most probable reason as discussed in Section 3.6.

Figure 9 compares the estimated naCO for two methods of calibration: one estimated through the difference of tCO and COOH (the canonical approach) and by direct calibration (alternate approach). We find that on average, the naCO is close to zero using both estimates (and within the differences of cCOH and tCO). For predictions with raw spectra, the range of predictions is smaller for naCO estimated directly than as a difference of cCOH and tCO. naCO estimates from PLS with
baseline corrected spectra are notably less variable than for those using raw spectra. Moreover, the canonical and alternate estimates are strongly correlated ($r = 0.99$ for urban and $r = 0.95$ for rural samples) even for these low concentrations, despite the fact that the two models use wavenumbers and latent variables differently (Figure S13). These results suggest that baseline correction can reduce interferences that may impart uncertainties in the estimation of FGs in this region for most samples, including clusters 7, 16, and 20.
High abundances of naCO have been reported in biomass burning and biogenic secondary OM in past studies (using PF), either due to ketones present in photochemical reaction products (Schwartz et al., 2010) or esterification in the condensed-phase (Russell et al., 2011). Therefore, it is surprising to find such abundances of naCO especially in rural sites with biogenic influences and samples influenced by residential wood burning (Kuzmiakova, *in prep.*, 2018). This finding may point to a differences in sample types between this and previous work, and possible artifacts due to long PM collection times, storage,
and transport protocols in monitoring network samples. For instance, more opportunities for conversion of naCO to aCOH by aldol condensation in the condensed phase may be possible in these samples.





## 3.5 Evaluation of estimated OM, OM/OC and O/C

Figure 10 (left column) summarizes estimates of OM, OM/OC, and O/C obtained by FTIR (distributions are shown in Figure S14). On average, concentrations of OM are higher in urban samples than rural ones, while the OM/OC ratio and O/C ratio shows the opposite pattern, as expected from previous studies. These trends are in agreement with measurements by GC-MS

and AMS (Turpin and Lim, 2001; Aiken et al., 2008), and in accordance with our understanding of atmospheric processes by which condensation of functionalized molecules (Ziemann, 2005; Kroll and Seinfeld, 2008) and heterogeneous reactions (Smith et al., 2009; Lim et al., 2010) lead to chemical aging.

The absolute magnitudes, however, require further consideration. The mean OM/OC values estimated by PLSr (Ruthenburg et al., 2014) of 1.5 and 1.6 for urban and rural sites, respectively, are within range of values previously reported by GC-MS and

AMS (Turpin and Lim, 2001; Aiken et al., 2008). However, the mean O/C ratio of 0.25 for rural sites is particularly low, and corresponds to values for hydrocarbon-like components derived from AMS PMF analysis (e.g., Aiken et al., 2008; de Gouw et al., 2009; Canagaratna et al., 2015). These results suggest that PLSr may be underestimating the oxygenated FGs (COOH and aCOH), especially for rural sites. The mean OM/OC ratio for PLSbc, PFo and PFr are higher than PLSr and range from 2.0 (urban) up to 2.1 (rural), with O/C ratios from 0.5 (urban) and 0.7 (rural). The surprisingly high values of OM/OC and O/C

for urban samples can be attributed to an understimation of FG OC, which inflates the OM/OC estimates. However, the mass fractions of FGs (shown as conventional pie graphs in Figure S10 of the supplementary material) suggests relative proportions estimated by PFo, PFr, and PLSbc are similar to urban aerosol composition previously reported by Russell and co-workers (Russell et al., 2011; Takahama et al., 2013a). The proportion of aCH mass is estimated above 40% and COOH and aCOH approximately one quarter each; primary amine and carbonyl comprise the rest of the average OM mass for the urban samples

(between 3 and 6%). In contrast, PLSr estimates the aCH fraction to be 71% for urban samples and 64% for rural (whereas the other models estimate between 35 and 40% for rural sites).

To improve these carbon-normalized metrics, the undetected carbon moeities can be corrected by incorporating an assumed carbon mass recovery fraction (Takahama and Ruggeri, 2017). Alternatively, an available OC reference measurement can instead be used for normalization — this can be TOR OC which we use here, or TOR-equivalent OC estimated from FTIR

spectra (Dillner and Takahama, 2015a; Reggente et al., 2016). This latter procedure leaves FTIR FG measurements to provide only the non-carbon atom content, which can be estimated with less uncertainty than the carbon content by using FG analysis (Section 2.6). The uncertainty in aCH abundance plays a critical role in estimation of carbon and OM mass, as nearly half of the total carbon is attributed to that associated with aCH. When using FG OC for normalization, contribution of this FG to the non-carbon portion of OM/OC is 0.17 at most (Appendix S2), but this belies the substantial role in governing the

overall magnitude of the ratio through its contribution to the OC estimate. However, if an external OC value is provided, the non-carbon contribution of aCH to OM is due to only hydrogen and the OM/OC (and O/C) is primarily dependent on estimates of the oxygenated fraction. Figure 10 (right column) summarizes estimates of OM, OM/OC, and O/C obtained by using FTIR estimates for non-carbon atom abundance, and TOR OC for carbon content. The adjustments reflect the extent of underestimation of TOR OC by each of the models, and, predictably, the PLSr mean rural OM/OC is reduced with respect to



its urban counterpart while the mean urban OM/OC is reduced with respect to the rural counterparts for the rest of the models. In the case of urban samples, the mean OM/OC varies between 1.5 (PLSr) and 1.8 (PFo), and the mean O/C varies between 0.20 (PLSr) and 0.48 (PFo). In the case of rural samples, the mean OM/OC varies between 1.5 (PLSr) and 2.0 (PFo), and the mean O/C varies between 0.21 (PLSr) and 0.61 (PFr).

When we heuristically adjust the PLSbc aCH model parameters to match TOR OC concentrations within 10% on average (PLSbc* introduced in Section 3.2), estimated OM, OM/OC, and O/C values fall within the extremes spanned by the various models. While laboratory calibrations can generate models that give reasonable predictions for ambient samples (to the extent that they can be evaluated), this comparison underscores the challenge in selecting the most appropriate model for ambient samples based on laboratory data. More experience in evaluating different model selection criterion on extrapolation is
necessary to improve the calibration strategy for FG estimation.

## 3.6   Anomalous samples

We examine and summarize in Figure 11 a few of the anomalous clusters. Cluster 7 (first row in Figure 11) samples have significant overprediction in both OC and ammonium for the calibration model using raw spectra (PLSr), but less in the baseline corrected models. These samples are found in almost all sites and are primarily influenced by dust (Kuzmiakova, *in prep.*,
2018). This conclusion is evidenced by the FTIR spectra having two sharp peaks above 3000 $cm^{-1}$ and a broader peak between 950 and 1100 $cm^{-1}$ indicative of Si-O bonds; resemblance of spectral features to hydroxyl groups from organic compounds or bound water in hydrates associated with dust are also observed. Accompanying XRF measurements also indicate high abundance of mineral dust elements in these samples. Larger atmospheric particles are likely to scatter infrared radiation, with increasing contributions above ∼200 nm (Signorell and Reid, 2010), and non-negligible contributions above one micrometer
(Allen and Palen, 1989). While the primary purpose of the baseline correction is to remove the scattering from the PTFE fibers, it is also likely that there is a scattering contribution from the particles which confers a positive artifact to the estimate of OC in the raw spectra calibration model. Baseline correction appears to reduce these artifacts through the removal of the particle scattering contribution to the observed absorbance.

Cluster 16 (second row in Figure 11) consists of wintertime Phoenix, AZ, samples which are associated with residential
woodburning (Kuzmiakova, *in prep.*, 2018). The consistent disagreement of the reference ammonium concentrations with all models suggests that the error may be attributed to the estimation of reference values rather than the calibrations. We expect gas/particle partitioning to favor the condensed-phase for ammonium nitrate for wintertime temperatures in Phoenix, so an evaporation artifact from Teflon is not anticipated to be the most significant factor. However, potassium nitrate is a well-known product of biomass burning, and the offset in ammonium equivalently formulated in the magnitude of potassium matches the
reported concentrations by X-Ray Fluorescence.

The atypical predictions for clusters 19 and 20 (third and forth row in Figure 11) are likely due to the abundance of large ammonium sulfate (cluster 19) and ammonium nitrate (cluster 20) particles in the samples, leading to anomalous transmission of infrared radiation (Christiansen peak effect) (Christiansen, 1885; Barnes and Bonner, 1936; Henry, 1948; Prost, 1973) through the sample (Figure S15). The Christiansen peak effect occurs under two limiting conditions: the refractive index




approaches that of the surrounding medium (air in this case), and the size of the particle(s) approach the wavelength of the incident radiation. The refractive index of ammonium sulfate and ammonium nitrate both exhibit a local minimum below 1.3 at 3.0 μm (3300 $cm^{-1}$) (Jarzembski et al., 2003). $PM_{2.5}$ can include some particles above 2.5 μm as the cutpoint corresponds to the median diameter of any particle efficiency curve of a size-selective inlet (cyclone for IMPROVE samples). However,

a another reason that the Christiansen effect plays a role in these samples is that its magnitude can still be significant for particles smaller than this diameter (Carlon, 1979). The result of this phenomenon is that the transmittance increases near this wavelength, though never approaching 100% on account of co-absorbing substances and inhomogeneities in atmospheric particles (Shelyubskii, 1993; Pollard et al., 2007). The corresponding absorbance spectrum displays a sharp decrease at the Christiansen wavelength relative to its neighboring absorbances and spectral distortions in its vicinity. This optical artifact can

affect both baseline correction and direct calibration (without baseline correction) if these effects are not taken into account, and our unexpected predictions can most certainly be attributed to this phenomenon. Remedies for this artifact may entail explicit modeling of the anomalous transmittance peak in the baseline correction or inclusion of samples which have this effect in the calibration samples. As both of these effects are nonlinear to absorbance, their treatment by a linear model may lead to a suboptimal representation of their contributions across multiple latent variables (including cross-over with contributions to

the signal such as instrument noise; Zupan and Gasteiger, 1991). Nonetheless, the demonstrated performance of calibration models for TOR-equivalent OC prepared from the regression of FTIR spectra to collocated TOR OC measurements (Dillner and Takahama, 2015a) suggests that PLS can handle these irregularities (scattering, Christiansen effect) as long as samples which exhibit them are included in the calibration samples, with or without baseline correction.

## 4  Conclusions

In this work, we explore the diversity in FG predictions that can result from calibration models built with mid-IR spectra. In particular, we compare two prominent methods for estimation of functional groups (FGs) from mid-IR spectra used in atmospheric PM analysis: peak fitting (PF) and partial least squares (PLS) regression. PF is an approach using physically-based absorption profiles to model spectral signals, and PLS is a statistical approach which is trained on relevant features from reference spectra. Using PF, we evaluated FG estimates using molar absorbance coefficients (model parameters) from previous

studies (PFo) and calculated (PFr) using 238 laboratory standards from Ruthenburg et al. (2014). Using PLS, we evaluated FG estimations using raw spectra (PLSr, in which substrate PTFE interferences are present) and baseline corrected spectra (PLSbc).

    PFo and PFr require some assumptions: (i) structure of the PTFE signal; (ii) value of the molar absorbance coefficients; (iii) apportionment rule to apportion carbonyl to carboxylic and non-acid contributions. Understimation of OC in comparison to

TOR (by as much as 50%) and surprisingly high values of OM/OC (greater than 1.8) for the urban site, Phoenix,is likely due to the understimation of aCH. Using a different value of the absorption coefficient, particularly for aCH, within uncertainty bounds presented in this study can mitigate this discrepancy. PLSr requires least prior knowledge — e.g., how to model the baseline — and therefore brings an appealing approach to calibration. However, scattering contributions from larger ambient





particles can lead to overprediction of organic FGs and ammonium in ambient samples with considerable dust impacts. As reported in previous studies (Ruthenburg et al., 2014; Takahama and Dillner, 2015) PLSr shows good agreement (correlation coefficients above 0.85 and regression slope close to 0.9) with external TOR OC measurements, especially for urban samples, and the OM/OC values are also within range of expected values (1.4–1.8). However, the higher values of OM/OC at the rural

sites may be due to an underestimation of the oxygenated FGs, which leads to a lower estimate of carbon content and an artificial increase the OM/OC. This bias is apparent when normalizing by TOR OC, as the OM/OC ratios of rural sites become similar to the urban values. PLSbc reveals the most consistent estimates against PFr, and this is sensible as the two use the closest correspondence of laboratory standards and spectral preparation (baseline correction).

Both PLSbc and PLSr can quantify carboxylic acid and non-acid carbonyl groups directly by designating the target variable

to COOH and naCO, and the models are trained on wavenumbers and LVs relevant for the two species. From this analysis, we conclude that almost all of the carbonyl for samples in the seven 2011 IMPROVE sites is associated with carboxylic rather than ketone or ester CO. In principle, it is also possible to define a fixed relationship between carboxylic cCOH and carbonyl CO such that COOH and the residual naCO can be determined in PF, but requires additional assumptions to implement.

In summary, models built with laboratory standards and algorithms are able to extract relevant information from ambient

FTIR sample spectra. Evaluation against external reference values (TOR OC and ammonium estimated from anion chromatography analysis) suggests moderately-strong to strong correlation for this IMPROVE monitoring data set, and is generally consistent with past that have also found high correlation with collocated measurements of TOR OC and AMS OM (e.g., Russell et al., 2009; Gilardoni et al., 2009; Takahama and Russell, 2011; Corrigan et al., 2013). However, the overall magnitude of bias can vary substantially depending on the choice of models and parameters. While we should also not expect perfect agreement

as each of the external measurements have their own artifacts, the sensitivity and resulting uncertainty in FG estimation due to available selection of parameters is apparent. Many parameters give validated predictions for laboratory standards, but each can give different results when applied to ambient samples (i.e., estimating concentrations in ambient samples is an ill-posed problem). Use of different absorption coefficients for PF and number of LVs for PLS that are still consistent within limits of the calibration standards in this work can offset apparent biases, but there are many parameters and their selection criteria are

at present time not sufficiently constrained. An example was shown where the overall magnitude in estimated FTIR OC can vary by 40% (and effectively eliminating bias with respect to TOR OC) by adjusting the number of LVs for aCH used by the PLSbc model.

Reducing uncertainty in predictions derived from FTIR spectra can be envisioned by two means: further advancing our study of laboratory standards that mimic ambient samples more closely, and by exploring mathematical solutions possible within a

stricter set of constraints. Regarding the first point, Takahama et al. (2016) notes that predictions from statistical calibration models (PLS and its variants) become less sensitive to model parameters as the samples in the calibration and prediction sets become more similar, and presumably this conclusion can be extended to the PF approach in its use of absorption profiles (both in intensity and shape). Given that some differences will remain between key features in laboratory standards and ambient samples, the second point on algorithmic improvements can be formulated in several ways. One strategy is to explore the

subset of solutions that are consistent with available external measurements (e.g., TOR OC, AMS OM, and other chemical



information) to revise model selection criteria. While an example varying the number of LVs for aCH is shown in this work, a more formal approach to multi-parameter optimization is preferable for approaching this task. Targeting means to establish different relationships between spectra and FGs than considered in this work is also possible. For instance, the full range of available calibration samples or absorption coefficients are likely not the most appropriate for every sample. Diversity in sample composition — e.g., between urban and rural samples — can be incorporated in a multilevel modeling approach, whereby different model or model parameters can be used based on spectral shape or identified source contributions (e.g., using Positive Matrix Factorization; Paatero and Tapper, 1994). Furthermore, models can be constrained to share a common representation to follow actual structure/spectra correlations more closely than when models for each FG are constructed independently — i.e., constraints on the internal representation of interferences toward organic FG quantification can be improved by concurrently developing our capability to model ammonium nitrate and ammonium sulfate using their discriminating bands. The same parameters (either absorption profiles in PF or LVs in PLS) that are able to accurately predict concentrations of these inorganic compounds would likely be able to model their interferences to organic FG absorption more correctly over a broader range of instances. Finally, anticipating the mass fraction of OC that can be explained by FGs will continue to play an important role in estimating the overall OM, and particularly for the OM/OC ratio. There are few classes of carbon atoms in molecules that are not expected to be detected by FTIR (i.e., they are not associated with FGs for which calibrations are not built), and understanding the expected extent of underestimation of OC by FG reconstruction will provide better perspective on evaluation of FG OC by TOR OC, and model selection methods. While the exact molecules in the aerosol mixture need not be enumerated for this purpose, knowledge of the functionalized carbon types that are present in different sample types are useful in this regard, and these can be obtained through measurements and simulation (Takahama and Ruggeri, 2017). In the meantime, continuing improvement in estimation of TOR-equivalent OC from direct calibration to collocated measurements (Dillner and Takahama, 2015a; Weakley et al., 2016) can enable estimation of carbon content from the same FTIR spectrum without imposing uncertainty from FG calibrations or requiring collocated TOR measurements.

FTIR spectroscopy remains a promising analytical technique to provide independent estimates of OM, OM/OC, and O/C based on molecular structure. At present time, a large number of calibration models can be generated based on selection of laboratory standards and algorithms, but further research is needed to develop a robust model selection process to reduce uncertainty in prediction when applied to ambient samples. Users should therefore note the existence of potential biases in current FTIR calibration models due to model or parameter sensitivty when performing comparisons against external measurements, while further progress is made toward development of calibration strategies.

## 5 Code availability

Baseline correction (Kuzmiakova et al., 2016), Peak-fitting (Takahama et al., 2013b) and multivariate calibration (Dillner and Takahama, 2015a, b) are implemented in an open platform with browser-interface, accessible at http://airspec.epfl.ch. Access to the software and their repositories are described in a companion manuscript.



## 6    Data availability

The IMPROVE network spectra will be made publicly available. TOR OC and PM$_{2.5}$ can be downloaded from the US Federal Land Manager Environmental Database at http://views.cira.colostate.edu/fed/DataWizard/Default.aspx.

*Competing interests.*    The authors declare that they have no conflict of interest.

5    *Acknowledgements.*    The authors acknowledge funding from EPFL and the IMPROVE program (National Park Service cooperative agreement P11AC91045). We also thank Christophe Delval for assistance in identification of the Christiansen peak species.



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





# Tables

**Table 1.** Summary of models evaluated.

| Label | Calibration data | Baseline correction | Algorithm | Model selection method[*] |
|---|---|---|---|---|
| *PLSr* | [1] | none (raw spectra) | PLS | randomization test |
| *PLSbc* | [1] | spline | PLS | randomization test |
| *PFo* | [2] | spline | peak fitting | arithmetic mean |
| *PFr* | [1] | spline | peak fitting | weighted mean |

| [1] | Ruthenburg et al. (2014) |
|---|---|
| [2] | Gilardoni et al. (2007), Russell et al. (2010), and Takahama et al. (2013b) |
| [*] | For PLS, method for selecting number of LVs for each FG; for peak fitting, method for selecting an absorption coefficient for each FG. |

**Table 2.** Recalibrated absorption coefficients and fit statistics for each FG and compound. Italicized texts denote the compounds not used in the computation of cCOH and aCH absorption coefficients averages.

| category | n | aCOH | cCOH | aCH | CO | NH |
|---|---|---|---|---|---|---|
| | | (abs. coef.±std. err., $R^2$) | | | | |
| 1-Docosanol | 3 | (37.7±2.0, 0.99) | – | (3.0±0.1, 1.00) | – | – |
| 1-Docosanol, Adipic Acid | 5 | (36.1±3.8, 0.96) | (51.63±4.88, 0.97) | (3.1±0.2, 0.99) | (13.66±0.26, 1.00) | – |
| 1-Docosanol, Suberic Acid | 5 | (29.8±0.6, 1.00) | *(0.0±0.0, 0.74)* | (2.6±0.1, 1.00) | (16.1±0.9, 0.99) | – |
| 1-Docosanol, Adipic Acid, Suberic Acid | 4 | (33.6±1.5, 0.99) | *(0.0±0.0, 0.36)* | (2.7±0.1, 1.00) | (15.9±1.4, 0.98) | – |
| 12-Tricosanone | 26 | – | – | (2.2±0.0, 1.00) | (11.0±0.1, 1.00) | – |
| Arachidyl Dodecanoate | 17 | – | – | (1.8±0.1, 0.99) | (10.2±0.5, 0.97) | – |
| D-Glucose | 5 | (23.9±0.6, 1.00) | – | (1.8±0.1, 1.00) | – | – |
| Fructose | 11 | (21.4±0.4, 1.00) | – | (1.6±0.1, 0.98) | – | – |
| Levoglucosan | 23 | (19.8±0.2, 1.00) | – | (0.8±0.0, 0.98) | – | – |
| Malonic Acid | 8 | – | (32.8±3.0, 0.95) | *(7.6±0.4, 0.98)* | (9.9±0.9, 0.95) | – |
| Suberic Acid | 17 | – | (41.3±0.9, 0.99) | (1.1±0.0, 0.99) | (13.7±0.1, 1.00) | – |
| Ammonium sulfate | 32 | – | – | – | – | (14.8±0.1, 1.00) |



# Figures

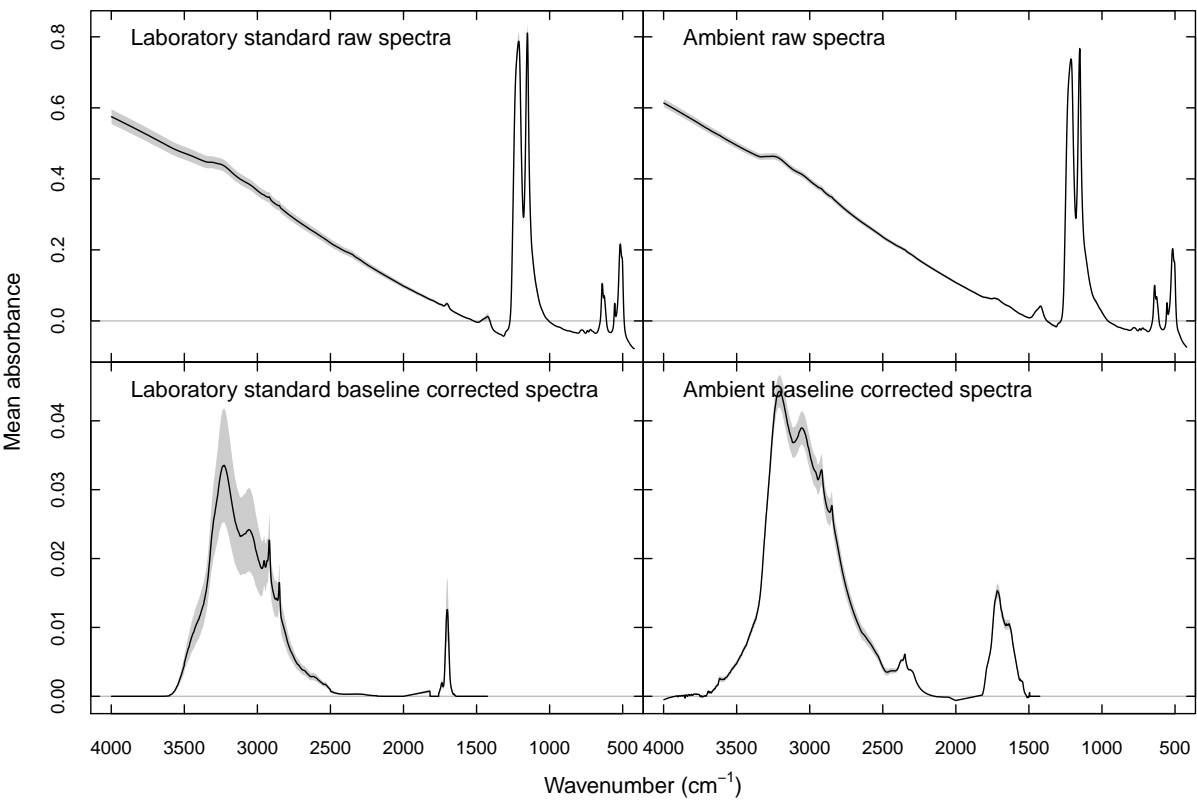

**Figure 1.** Laboratory and ambient sample spectra (raw and baseline corrected). Black lines denote mean absorbances, and dashed gray areas denote 95% confidence intervals.



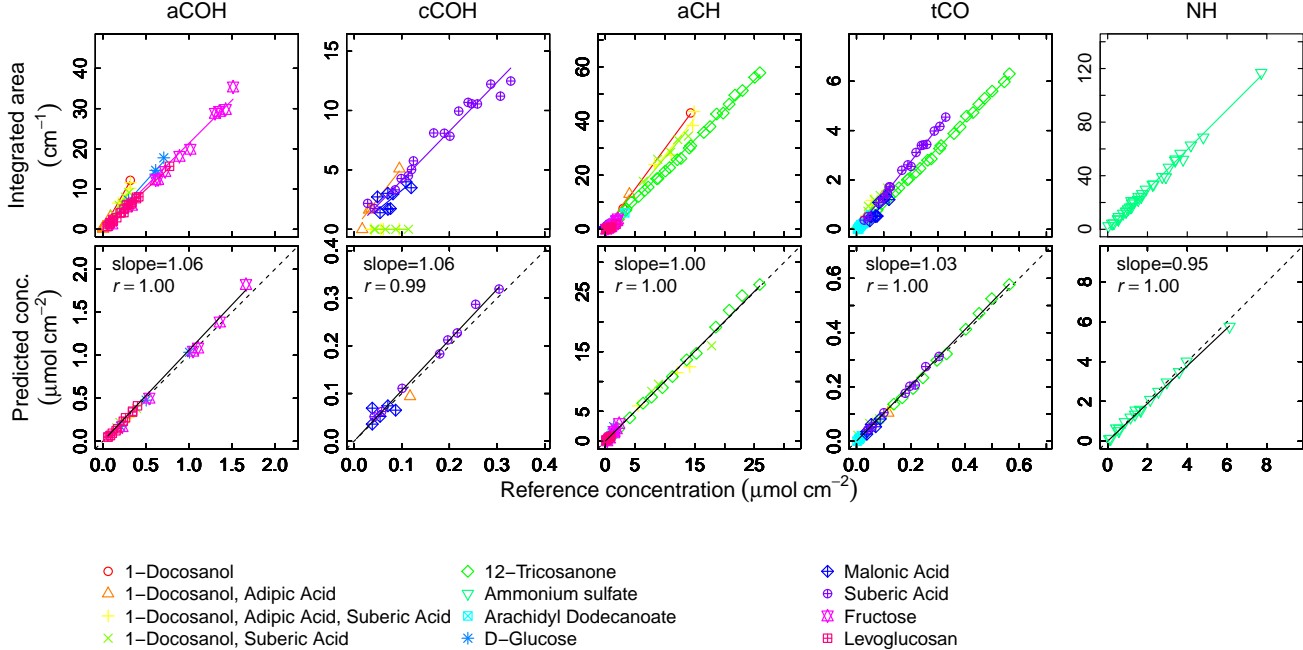

**Figure 2.** Top row: integrated absorption as a function of known molar abundance used to derive molar absorption coefficients. Bottom row: evaluation of derived absorption coefficients on predicted concentrations for test set compounds not used in the fitting.

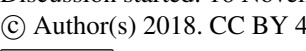





**Figure 3.** Summary of molar absorption coefficients reported in the literature. The single star for 1-docosanol aCOH indicates that there are only three points — one of which is an influential point — so this is effectively a single-point estimate. The single star for ammonium sulfate indicates that it based on a single value. The double star is used to indicate that the absorption coefficient for malonic acid cCOH is estimated for a concentration range order of magnitude lower than the rest. Previous studies are summarized by Takahama et al. (2013b), which also compiles coefficients from Gilardoni et al. (2007) and Russell et al. (2010).




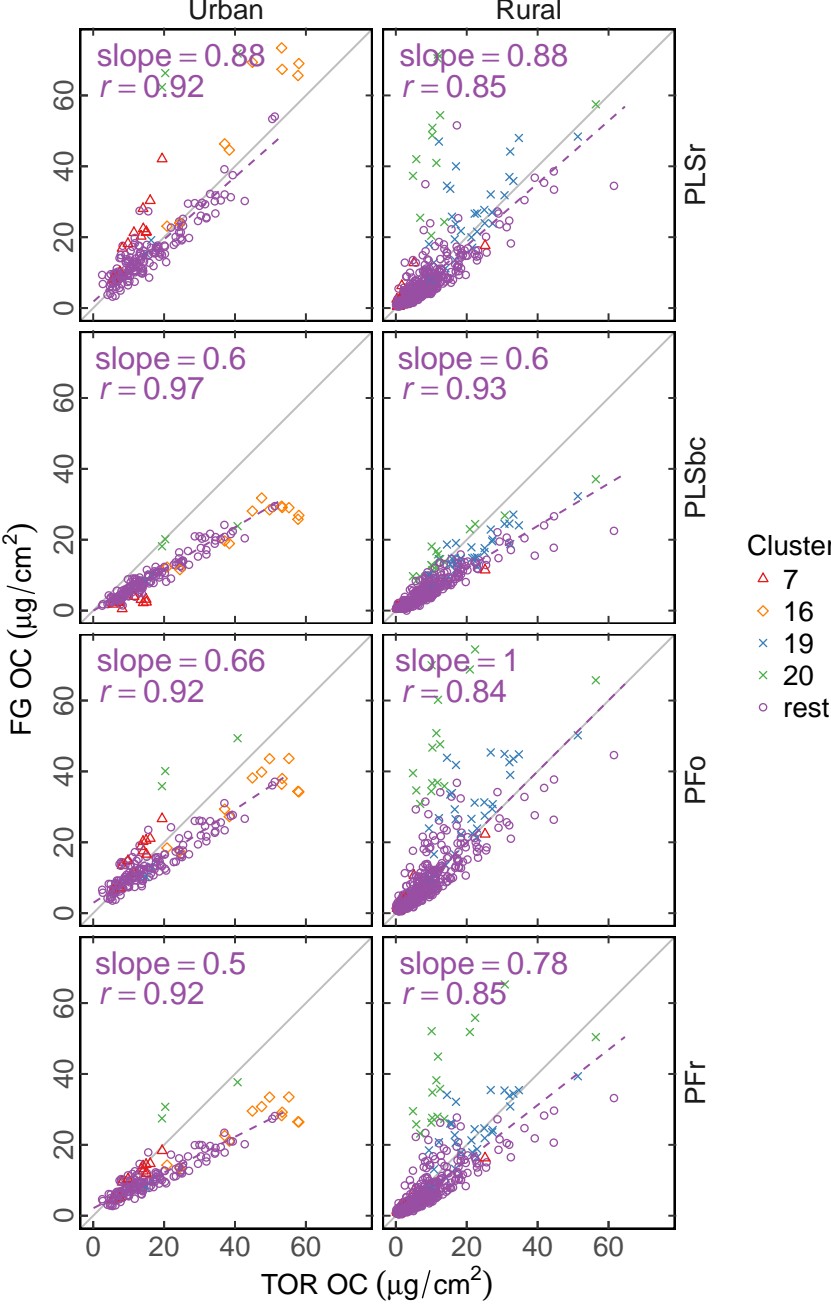

**Figure 4.** Comparison of estimated OC (FG OC) against OC measured by TOR method (TOR OC). PFo refers to peak-fitting using the original parameters. PLSr refers to partial least square using raw spectra. PFr refers to peak-fitting using the recalibrated absorption coefficients. PLSbc refers to partial least square using baseline corrected spectra.



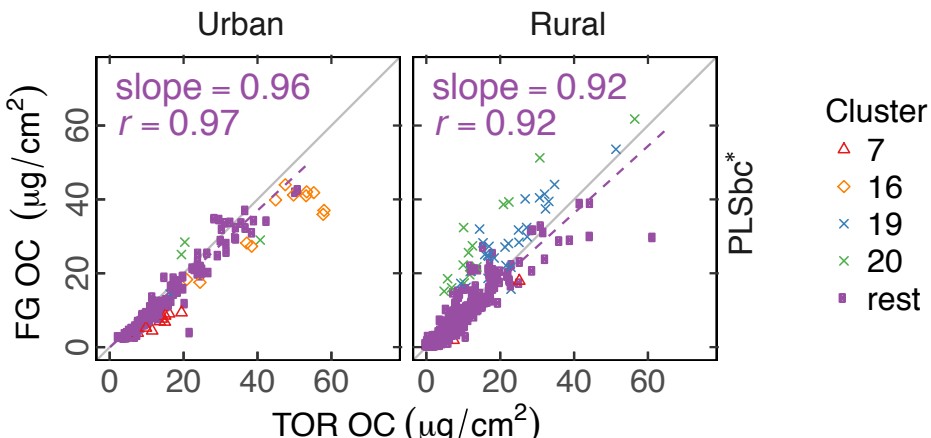

**Figure 5.** Comparison of estimated OC (FG OC) gainst OC measured by TOR method (TOR OC). PLSbc* refers to partial least square using baseline corrected spectra and heuristic choice for the aCH LVs number (13) based on agreement between FG OC with TOR OC (Figure S16).



**Figure 6.** Comparison of estimated Ammonium (FG Ammonium) against Ammonium measured using ion chromatography (IC Ammonium). PFo refers to peak-fitting using the original parameters. PLSr refers to partial least square using raw spectra. PFr refers to peak-fitting using the recalibrated absorption coefficients. PLSbc refers to partial least square using baseline corrected spectra.





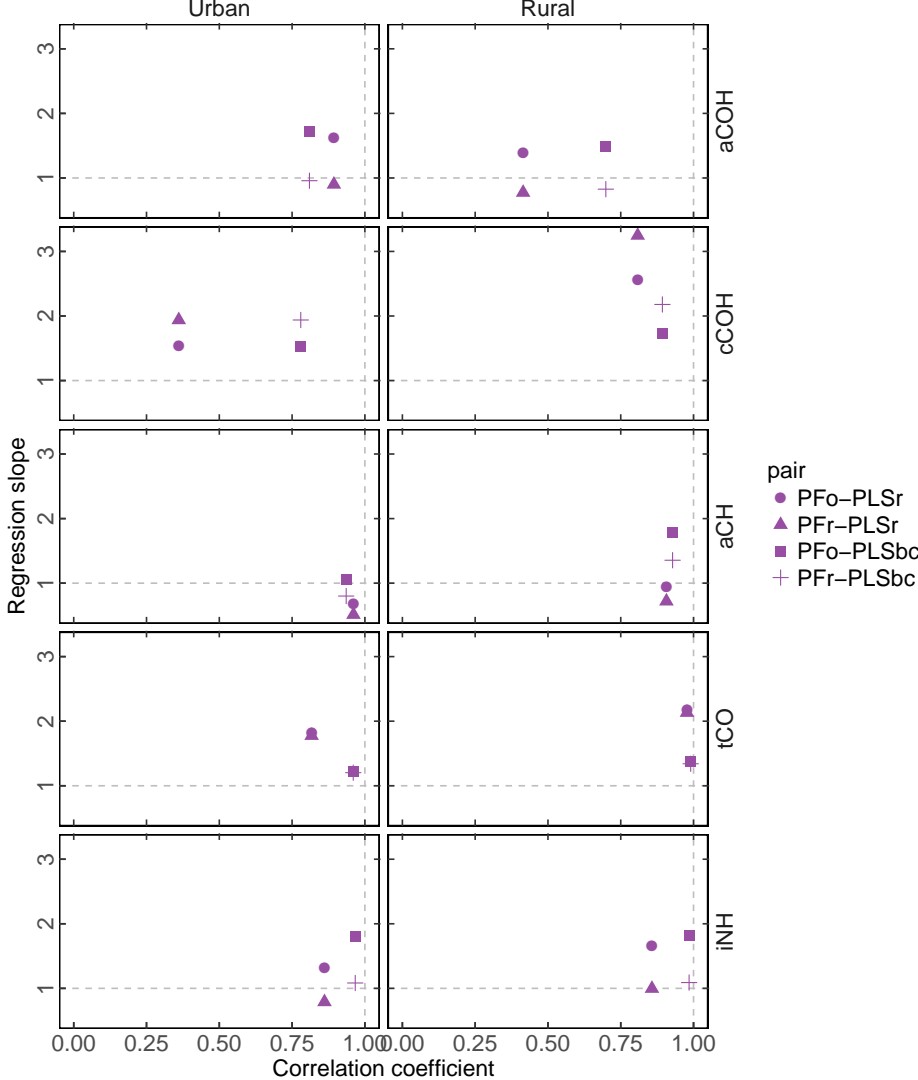

**Figure 7.** FG comparison summary. PFo refers to peak-fitting using the original parameters. PLSr refers to partial least square using raw spectra. PFr refers to peak-fitting using the recalibrated absorption coefficients. PLSbc refers to partial least square using baseline corrected spectra.




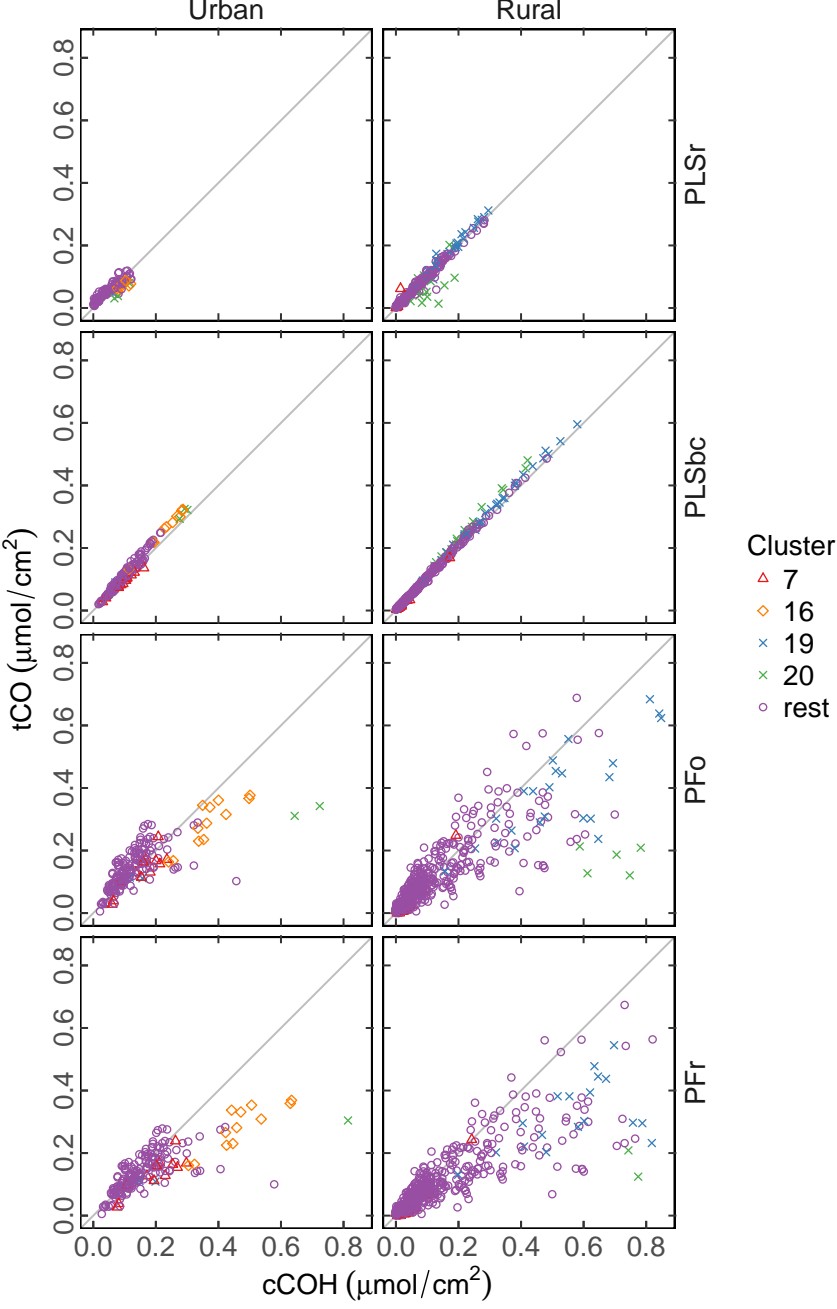

**Figure 8.** Comparison of quantified abundance of tCO and cCOH. PFo refers to peak-fitting using the original parameters. PLSr refers to partial least square using raw spectra. PFr refers to peak-fitting using the recalibrated absorption coefficients. PLSbc refers to partial least square using baseline corrected spectra.





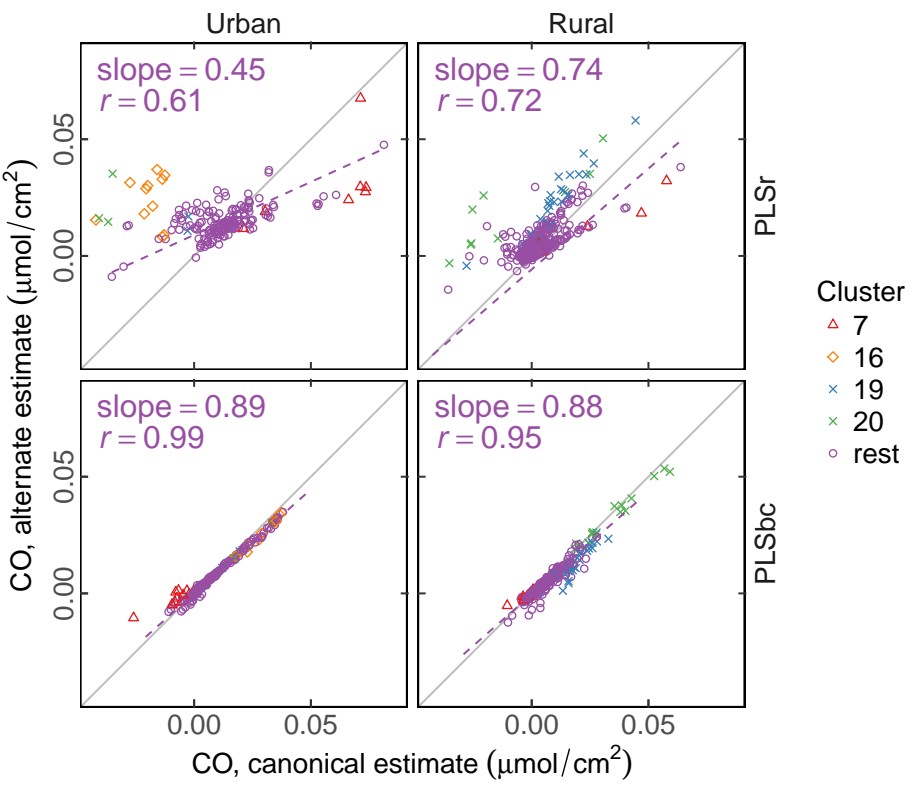

**Figure 9.** Comparison of estimated CO according to canonical calibration (as difference between molar abundance of cCOH and tCO), and alternate calibration (direct calibration to non-acid CO). PLSr refers to partial least square using raw spectra. PLSbc refers to partial least square using baseline corrected spectra.





**Figure 10.** Top row: barplots of OM mass fractions from quantified FGs. Middle row: barplots of OM/OC ratio and associated non-carbon atoms. Bottom row: O/C. In the left panel, OM, OM/OC and O/C ratios use OC estimated by FG calibrations (FG OC). In the right panel, the OM, OM/OC and O/C ratios use OC measured by TOR. PFo refers to peak-fitting using the original parameters. PLSr refers to partial least square using raw spectra. PFr refers to peak-fitting using the recalibrated absorption coefficients. PLSbc refers to partial least square using baseline corrected spectra. PLSbc* refers to partial least square using baseline corrected spectra is the same as PLSbc except that the number of LVs for aCH has been selected heuristically (Section 3.2.)




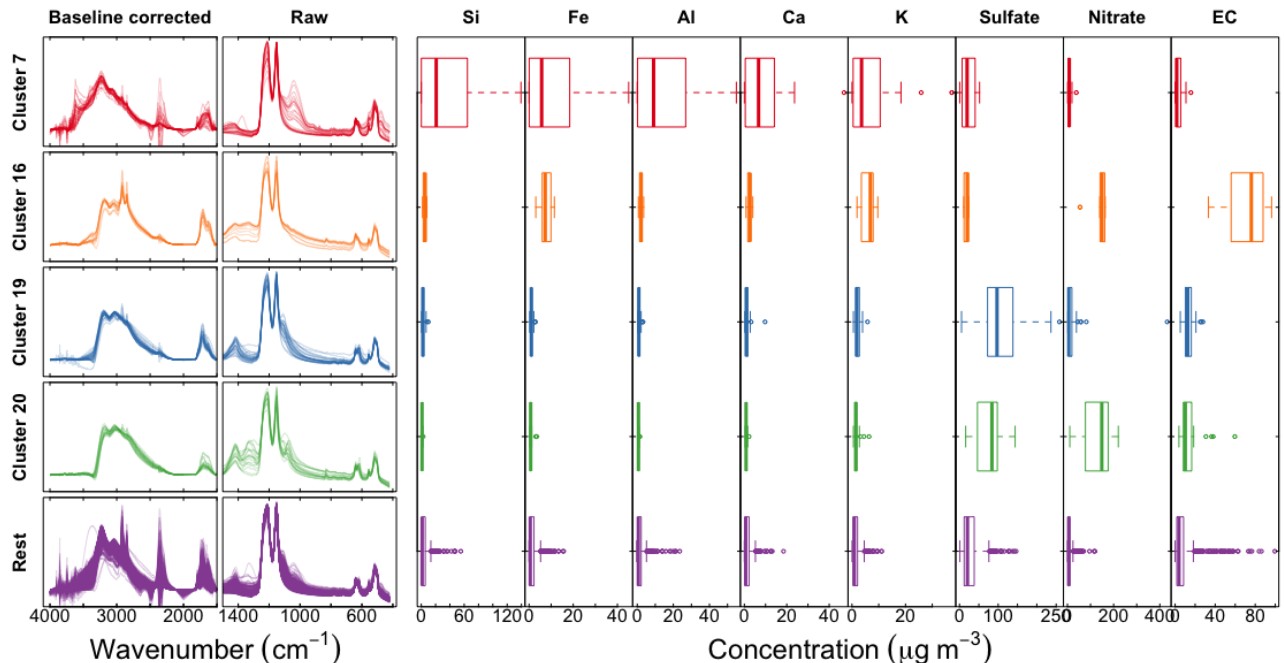

**Figure 11.** Left column shows scaled baseline corrected spectra between 4000 and 1500 $cm^{-1}$; middle column shows scaled raw spectra below 1500 $cm^{-1}$, and concentrations of PM constituents measured in the IMPROVE network: trace elements from X-Ray Fluorescence, inorganic ions (sulfate and nitrate) from ion chromotography, and elemental carbon from TOR analysis.

.