# Peer review of "Analysis of functional groups in atmospheric aerosols by infrared spectroscopy: systematic intercomparison of calibration methods for US measurement network samples"

_Atmospheric Measurement Techniques, 2018_

## Referee Comment (RC1) · Anonymous Referee #1 · 16 Dec 2018

Summary:

In this work, Reggente et al. compare two methods to derive functional group abundance from FTIR spectra of aerosol collected on filters. The methods compared are peak fitting (PF) and partial least squares regression (PLSR). Total Organic Carbon (OC) was validated against Thermal Optical Reflectivity (TOR-OC)–which separates organic mass from refractory black carbon. Also validated was the NH functional group concentrations against sulfate-plus-nitrate concentrations.

Overall, the general topic of this work is of interest to the Atmospheric Measurement Techniques community. Devolving complex spectra into their components is a powerful technique that is applicable beyond even FTIR. There are, however, several major and minor comments I have about this work, which are summarized below.

Major Comments:

My main major comment about this work is that much of it has been developed and validated in previous papers. The authors do not clearly outline this in their introduction or methods. In fact, readers are required to check at least three separate papers to fully grasp what is novel about this work. The three previous papers mentioned are as follows:

Takahama et al., 2013 developed and validated the peak fitting analysis for COH and CO.

Ruthenberg et al., 2014 analyzed the same dataset as this paper and ran PLS (raw) on the samples. They also validated their data using TOR-OC.

Kuzmiakova et al., 2016 developed background correction protocols for peak fitting analysis. They also tested the background-corrected peak fitting analysis against TOR-OC.

In summary, none of the data, functional group fitting techniques, or baseline corrections presented in this paper are novel. One new product is the updated molar absorption coefficients. Thus, the only real new analysis are the baseline corrected PLSR data. Conceivably, this work could be labelled as an intercomparison; however, that is difficult to justify for only two methods and no additional functional group verification. Ultimately, this paper would benefit for an explicit description of what has been done previously, and a similarly explicit description of how this paper improves upon previous techniques or instructs the field on how to best use these functional group analyses for filter-based FTIR spectra.

My second major comment is related to functional group verification. While the focus of this paper is determining the functional group abundance in US measurement networks–only one functional group validation (NH) was made. Ultimately, the authors conclude that "further progress in parameter selection [i.e., molar absorption coefficients for PF and number of latent variables for PLSR] is required." This belies the entire purpose of this article, which is to give solidarity to these techniques and their ability to estimate functional group abundances from filters collected in US networks. Perhaps the authors could assess the current literature, or find additional data products from the IMPROVE network other than TOR OC and ammonium, to help validate their measurements.

Finally, because this work relies heavily on methods used in previously published papers from this group, the authors often do not fully describe their techniques. While this is understandable for details of the technique, oftentimes entire concepts are missing. This forces the reader to read several other papers to grasp the main concepts of the techniques used in paper. As much as I could, I have tried to outline this in the minor comments section; however, the authors should also pay close attention to this–especially when describing both the specifics of PF algorithms and PLSR in the context of identifying functional groups in FTIR spectra.

Minor Comments:

P3L9-17: There are 32 (!) references in these two sentences. Ultimately, it is more distracting than helpful. I suggest keeping one or two of the most representative references for each functional group/type.

P3L23: This apportionment of detectable vs non-detectable fractions is hard to follow here. It might be more instructive to provide an example. While the Takahama and Ruggeri 2017 paper is referenced, this work should, at least conceptually, stand on its own.

P3L33: Is it not true that the peak shapes for single or simple components are also

Gaussian? Certainly, you must fit Gaussians to your simple components for calibration.

P4L34: The objective of this paper is to evaluate the robustness in estimated abundances. What is your measure of robustness–can you quantify this?

P8L4: It seems slightly confusing to me to use n_ik(a) here as the number of moles of bond n, since it was used as areal surface density earlier.

P8L17: This "apportioning" method needs to be described explicitly. As of now, the reader must to go the Takahama et al., 2013b reference to even understand the concept of the method.

P8L31 (Figure 3): I am not certain how you are calculating the error bars on your absorption coefficients. Just by eye, both the errors of the individual components and the assumed standard deviation of all the samples looks to be much larger than what you reported.

P9L29: How are you quantifying "significantly worse?"

P12L2: It might be clearer to remind the readers that the calibration curves are from 158 of the 238 laboratory standards and are used to derive a calibrated absorption coefficient. Then, the remaining laboratory standards are used in the predicted concentrations to validate the calibrated absorption coefficients. This is largely omitted from the text of this paragraph and requires the reader to rely only upon Figure 2 to interpret these crucial results.

P12L13: This paragraph would be clearer if the authors explicitly outline why they decided to re-run the molar absorption, and how it differs from both Russell and coworkers and Takahama et al., 2013. A difference is prescribed by the authors to a different baseline correction, but there are differences between just Russel and coworkers and Takahama et al., 2013.

P13L3: This should be a main result of this paper; however, as mentioned in the main comments, there are no ways to validate the functional group molar densities;

[Figure]

therefore, there is no way to validate the molar absorption coefficients. This calls into serious question the utility of this work.

P13L26. Could it also be that both are generally underpredicting by 40% or more, but the PLSr has some erroneous contribution from the PTFE filter? More discussion should be added here as PLSbc is one of the new techniques introduced to this paper.

P14L3: Is there some sort of tradeoff for not using PLSbc*? The authors mentioned overfitting leading to unrealistic results in their methods section–but it seems like the PLSbc* results shown here are more realistic.

P14L32: Is there any direct evidence that PLS2 outperforms PLS1 for identification of aerosol functional groups in FTIR?

Section 3.5: Again, these are useful comparisons–but without an external validation method, it is difficult to choose which methods are most robust. For example, PLSr seems to be doing well in predicting TOR-OC (Figure 2) and potentially OM/OC, but its OC value is low. The other methods have low TOR OC and possibly too high OM/OC, but realistic O/C. Which method are we to trust as the reader? Which method is most robust? Additionally, the TOR-OC normalized results seem more realistic–but they are hard to evaluate without external validation.

Section 3.6: This entire section, while interesting, seems like it would be better suited to go into the Supporting Information.

Technical Comments:

P1L2: I'm not sure that FTIR needs to be defined, as it is commonly known acronym.

P1L7: The phrase "including their model parameters" is parenthetical and could be enclosed by commas.

P7L9: OC is used here, but has not been previously defined.

P1L11: It would be much more constructive to explain what the "series of possible

input parameters" refers to. As it reads now, it is not instructive and, therefore, is unnecessary.

P1L14: TOR OC is used here, but it has not been defined.

P2L9: OM is used here, but it has not been previously defined.

P2L27: Like the FTIR calibration, readers should be generally familiar with the most general principle of FTIR. This could be cut for brevity.

P3L23: Should FGss be FGs?

P4L16: PLS and PF "have," not "has"

P7L6: There should be a space before "A single parameter ..."

P10L16: The phrase "is depends" should be just "depends"

P14L24: Should the phrase read "no gross overestimate, or that underestimation is unlikely?"

p19L5: There is an extra "a" at the beginning of this line.

p19L30: There should be a space after "Phoenix," and before "is." Figure 10. The models do not need to be defined here as they are defined earlier and used often since.

Figure 2. It might be clearer to use different colors in the bottom row to highlight the fact that you've used the test samples to validate the calibration made with the calibration samples. Figure 7. The urban x-axis value on the right side is overlapping with the rural x-axis on the left side. Also, is the correlation coefficient shown here R or R^2?

Figure 10. This figure would be much clearer if you centered the descriptions at the top (e.g., "From FG calibrations") and made them larger or bold.

---

## Referee Comment (RC2) · Anonymous Referee #2 · 19 Dec 2018

General comments

This manuscript details the similarities and differences between peak fitting and partial least squares as means of interpreting the organic functional group composition of ambient aerosol. The work is appropriate for AMT and relevant to the scientific community as filter samples are a routine way to collect atmospheric particles. Further FTIR is a non-destructive technique, so this method can be paired with other methods of analyzing filter samples. This work builds upon recently published advances in the area of FTIR spectroscopy. The paper presents the quantitative comparison of two

methods and provides sufficient detail for readers to understand the merits of partial least squares as an approach to FTIR analyses. All assumptions are clearly explained. In many cases, sensitivity type analyses are offered as well. The abstract provides a good summary of the article itself. The figures are of high quality and the table and figures are easy to read and understand. With the incorporation of minor suggested changes below, I support this manuscript for publication.

Specific comments

The title is satisfactory as written but does not indicate that the paper is primarily about the method of functional group quantification and not about the results themselves.

The manuscript is somewhat difficult to read, likely owing to the use of many acronyms and exhaustive detail of methods and results. In many cases, the authors could improve the readability of the manuscript by occasionally using the unabbreviated term (PLS, LVs, PF, etc) before returning to abbreviations.

Technical corrections

Pg 1 line 14: TOR used without definition

Pg 2 line 5: use "compose" instead of "comprise"

Pg 2 line 7: "methods . . .. include"

Pg 2, line 26: insert "and" before "ion chromatography"

Pg 11 line 15: I'm not sure if it should be "correspond" or "corresponds" please check the sentence meaning

Pg 11 line 31: I'm not sure what "evaluations of estimated quantities of using these absorption coefficients" means

Pg 13 line 23: define or redefine PFo and PFr

Pg 13 line 26: define or redefine PLSr and PLSbc

[Figure]

Pg 14 line 12: missing word "are thought to be an upper bound"

Pg 14 lines 13-14: awkward sentence "no evaporation loss of ammonium nitrate from PTFE or nitrate association with dust instead of ammonium are not considered"

Pg 14 line 25: do the authors mean "is likely"? Otherwise it is a double-negative

Pg 15 line 20: use "compose" instead of "comprise"

Pg 15 line 22: "vary" not "varies"

Pg 15 line 23: "correlation . . . is"

Pg 15 line 25: coefficient should be plural as it refers to two (14.84 and 8.89)

Pg 15 line 26: "correlation . . . is"

Pg 16 line 4: what is areal density?

Pg 16 line 27: I think the authors mean "such low abundances"

Pg 19 line 5: omit "a" before "another"

Pg 19 line 30: insert a space between "Phoenix, is"

---

## Author Comment (AC1) · 9 Mar 2019

**Response to reviewer comments: "Analysis of functional groups in atmospheric aerosols by infrared spectroscopy: functional group quantification in US measurement networks"**

We thank the editor and reviewers for feedback on the manuscript. We have addressed each comment below with responses in blue font, and submitted revised manuscript (with new additions highlighted in red font and old text for deletion in gray).

**Reviewer 1**

In this work, Reggente et al. compare two methods to derive functional group abundance from FTIR spectra of aerosol collected on filters. The methods compared are peak fitting (PF) and partial least squares regression (PLSR). Total Organic Carbon (OC) was validated against Thermal Optical Reflectivity (TOR-OC)–which separates organic mass from refractory black carbon. Also validated was the NH functional group concentrations against sulfate-plus-nitrate concentrations.

Overall, the general topic of this work is of interest to the Atmospheric Measurement Techniques community. Devolving complex spectra into their components is a powerful technique that is applicable beyond even FTIR. There are, however, several major and minor comments I have about this work, which are summarized below.

Major Comments:

- My main major comment about this work is that much of it has been developed and validated in previous papers. The authors do not clearly outline this in their introduction or methods. In fact, readers are required to check at least three separate papers to fully grasp what is novel about this work. The three previous papers mentioned are as follows:

  - Takahama et al., 2013 developed and validated the peak fitting analysis for COH and CO.

  - Ruthenberg et al., 2014 analyzed the same dataset as this paper and ran PLS (raw) on the samples. They also validated their data using TOR-OC.

  - Kuzmiakova et al., 2016 developed background correction protocols for peak fitting analysis. They also tested the background-corrected peak fitting analysis against TOR- OC.

In summary, none of the data, functional group fitting techniques, or baseline corrections presented in this paper are novel. One new product is the updated molar absorption coefficients. Thus, the only real new analysis are the baseline corrected PLSR data. Conceivably, this work could be labelled as an intercomparison; however, that is difficult to justify for only two methods and no additional functional group verification. Ultimately, this paper would benefit for an explicit description of what has been done previously, and a similarly explicit description of how this paper improves upon previous techniques or instructs the field on how to best use these functional group analyses for filter-based FTIR spectra.

> We thank the reviewer for this perspective. The point of this manuscript is indeed more of a systematic intercomparison of models that result from several key decisions - calibration data, baseline correction, and regression algorithm - applied to the same data set; only one of which has been applied to the IMPROVE 2011 data set previously by Ruthenburg et al. 2014. Furthermore, the most extensive set of evaluations for organic aerosol characterization to date has been conducted on submicron samples compared to aerosol mass spectrometry measurements (discussed now in the introduction); the current manuscript has identified "anomalous" predictions are likely attributable to Christiansen peak effects that can occur in PM2.5 samples with high dust particle loadings. We have changed the title to reflect this goal:

"Analysis of functional groups in atmospheric aerosols by infrared spectroscopy: systematic intercomparison of calibration methods for US measurement network samples"

We have furthermore highlighted critical needs in the field to better motivate the work presented here:

[revised manuscript text omitted]

- My second major comment is related to functional group verification. While the focus of this paper is determining the functional group abundance in US measurement networks — only one functional group validation (NH) was made. Ultimately, the authors conclude that "further progress in parameter selection [i.e., molar absorption coefficients for PF and number of latent variables for PLSR] is required." This belies the entire purpose of this article, which is to give solidarity to these techniques and their ability to estimate functional group abundances from filters collected in US networks. Perhaps the authors could assess the current literature, or find additional data products from the IMPROVE network other than TOR OC and ammonium, to help validate their measurements.

> We regret that the expectations for this manuscript was set too high; it is not yet at a point where we can claim solidarity among methods. Compared to other techniques, the development and evaluation for FTIR analysis has been quite limited thus far, especially considering the number of various vibrational modes from which information can be extracted. We have therefore changed the title and the introduction to highlight the status of quantification and how this paper contributes to the current state-of-the-art. (Modification to text included in response to comment above.)

> As the first systematic intercomparison of ways to interpret the same spectra, the need for further evaluation to select the best model does emerge from this work. Unfortunately, there are no other metrics to directly evaluate functional group composition in the monitoring network. Therefore, we propose for further work that evaluation against other methods (mass spectrometry, spectrophotometry) be conducted in limited field campaigns or laboratory studies, and evaluations for organic functional group quantification be conducted jointly with its spectral interferents, such as ammonium.

> In the Introduction, we have now included a discussion of past evaluations on different data sets so that where this manuscript fits in is better framed within the current state of research. The outcome of this article is to identify results which different algorithms have in common and those in which predictions diverge to identify areas which require further constraints.

- Finally, because this work relies heavily on methods used in previously published papers from this group, the authors often do not fully describe their techniques. While this is understandable for details of the technique, oftentimes entire concepts are missing. This forces the reader to read several other papers to grasp the main concepts of the techniques used in paper. As much as I could, I have tried to outline this in the minor comments section; however, the authors should also pay close attention to this–especially when describing both the specifics of PF algorithms and PLSR in the context of identifying functional groups in FTIR spectra.

> We have tried to highlight the key pieces of information which underscore differences among the algorithms and focus the bulk of our manuscript on the results. We have tried to clear up any confusion or ambiguities as suggested below.

Minor comments:

- P3L9-17: There are 32 (!) references in these two sentences. Ultimately, it is more distracting than helpful. I suggest keeping one or two of the most representative references for each functional group/type.

> We have substantially edited this section to focus on previous evaluations and modifications can be found in the revised manuscript. (Response to first major comment.)

- P3L23: This apportionment of detectable vs non-detectable fractions is hard to follow here. It might be more instructive to provide an example. While the Takahama and Ruggeri 2017 paper is referenced, this work should, at least conceptually, stand on its own.

> We have added the statement to the Introduction:

> "For instance, carbon atoms with aliphatic CH groups are detectable if CH is included in the suite of calibrations, while skeletal carbon atoms bonded only to other carbon atoms are considered undetectable."

- P3L33: Is it not true that the peak shapes for single or simple components are also Gaussian? Certainly, you must fit Gaussians to your simple components for calibration.

> We do indeed use the same Gaussian peak specifications are used for fitting laboratory and ambient sample spectra. We now also state that Gaussians are used for fitting laboratory and ambient samples in the Methods section.

- P4L34: The objective of this paper is to evaluate the robustness in estimated abundances. What is your measure of robustness — can you quantify this?

> We characterize robustness using two measures - the correlation coefficient characterizes the consistency between predictions, and the slope of a total least squares regression line characterizes

the magnitude of differences. We have added the statement: "Values estimated with slopes and $r$ close to unity among different methods are considered more robust." to the Methods section.

- P8L4: It seems slightly confusing to me to use n_ik(a) here as the number of moles of bond n, since it was used as areal surface density earlier.

    Thank you for catching this error - the two represent the same quantity (moles per unit area) and this has been revised in the text (Methods section).

- P8L17: This "apportioning" method needs to be described explicitly. As of now, the reader must to go the Takahama et al., 2013b reference to even understand the concept of the method.

    We have changed:

    "The apportionment protocol assumes that all FGs are present in each sample, which is a convenient approximation in atmospheric samples; for laboratory standards or specific source samples, the FGs to be fitted is specified for each spectrum"

    to:

    "The apportionment protocol is based on initial values and constraints set out by analysis of a large number of laboratory and ambient samples as described by Takahama et al. (2013). While peaks for FGs in laboratory standards only include those present in the compound, all FGs are assumed to be present in each sample, which is a convenient approximation in atmospheric samples."

- P8L31 (Figure 3): I am not certain how you are calculating the error bars on your absorption coefficients. Just by eye, both the errors of the individual components and the assumed standard deviation of all the samples looks to be much larger than what you reported.

    The error bars represent plus/minus one standard error of the absorption coefficients, and we used the same values reported in Table 3. We have added in Fig. 3 caption the following sentence:

    "The error bars represent plus/minus one standard error of the absorption coefficients".

- P9L29: How are you quantifying "significantly worse?"

    To clear up the ambiguous wording, we have rephrased this sentence to read: "This method selects a model with fewer LVs for which the squared prediction error is not statistically greater than the reference (minimum RMSECV) model."

- P12L2: It might be clearer to remind the readers that the calibration curves are from 158 of the 238 laboratory standards and are used to derive a calibrated absorption coefficient. Then, the remaining laboratory standards are used in the predicted concentrations to validate the calibrated absorption coefficients. This is largely omitted from the text of this paragraph and requires the reader to rely only upon Figure 2 to interpret these crucial results.

    We have changed: "Calibration curves and predicted concentrations according to the peak-fitting strategy outlined in Section ?? are shown in Figure 2. Regression parameters are included in Table 2."

    to:

    "Calibration curves and predicted concentrations according to the peak-fitting strategy outlined in Section ?? are shown in Figure 2. The top row refers to the calibration curves to compute the absorption coefficients (obtained using two-thirds of the 238 laboratory standards), and the bottom row refers to the evaluation of derived absorption coefficients on predicted concentrations for test set compounds not used in the calibration. Regression parameters (including the number of samples used in each category, n are included in Table 2."

- P12L13: This paragraph would be clearer if the authors explicitly outline why they decided to re-run the molar absorption, and how it differs from both Russell and coworkers and Takahama et al., 2013. A difference is prescribed by the authors to a different baseline correction, but there are differences between just Russell and coworkers and Takahama et al., 2013.

 analyzed the spectra manually, and therefore the difference is attributed to baseline correction and fitting. Since Russell et al. (2009a), all analysis using the peak fitting approach has followed the same protocol and code base (a refactored version is distributed with a companion manuscript); therefore, the differences are attributed to choice of molecule.

- P13L3: This should be a main result of this paper; however, as mentioned in the main comments, there are no ways to validate the functional group molar densities; therefore, there is no way to validate the molar absorption coefficients. This calls into serious question the utility of this work.

  While molar absorption coefficients for each compound can be validated, for complex mixtures comprising atmospheric particles the objective is to find a representative value of molar absorption coefficients that are consistent with available information (e.g., other measurements), which can give us more confidence in the functional group distribution. The results from FTIR at this time can be viewed as a data product to be continually revised to provide useful information. A more expanded set of compounds and variability in absorption coefficients, and evaluation against ambient OC will be presented in a forthcoming manuscript.

- P13L26. Could it also be that both are generally underpredicting by 40% or more, but the PLSr has some erroneous contribution from the PTFE filter? More discussion should be added here as PLSbc is one of the new techniques introduced to this paper.

  To a first estimate, we expect that the contributions from PTFE are minimal, as predictions for laboratory standards sampled on the same type of PTFE filters show excellent agreement with reference concentrations. We have added this statement to the text: "As FG abundances in laboratory samples are reproduced with minimal error, we anticipate that PTFE interferences to predictions are minimal with PLSr. [...]".

- P14L3: Is there some sort of tradeoff for not using PLSbc*? The authors mentioned overfitting leading to unrealistic results in their methods section — but it seems like the PLSbc* results shown here are more realistic.

  PLSbc* is only one of many possible results that show improved agreement with TOR OC; for this manuscript we have restricted our main comparisons to models constructed using only knowledge obtained with laboratory data according to previous calibration strategies (as this is the ultimate goal). This work brings to the community's attention that additional work is needed to understand and improve out FG models. In a separate manuscript (in preparation), we are exploring methods for constraining model parameters according to collocated ambient measurements.

  We have added the statement in the main text: "PLSbc* is only one of many possible models that show improved agreement with TOR OC to be explored in future work; for this paper we restrict restrict our evaluation of results primarily to those obtained by the protocols described in Section 2."

- P14L32: Is there any direct evidence that PLS2 outperforms PLS1 for identification of aerosol functional groups in FTIR?

  PLS2 and PLS1 has not been extensively compared in the literature, but we draw this inference from the framework of multi-task learning where more robust models can be constructed when subjected under additional constraints imposed under different contexts.

- Section 3.5: Again, these are useful comparisons — but without an external validation method, it is difficult to choose which methods are most robust. For example, PLSr seems to be doing well in predicting TOR-OC (Figure 2) and potentially OM/OC, but its OC value is low. The other methods have low TOR OC and possibly too high OM/OC, but realistic O/C. Which method are we to trust as the reader? Which method is most robust? Additionally, the TOR-OC normalized results seem more realistic–but they are hard to evaluate without external validation.

  As the author is likely aware, many environmental predictions cannot be independently validated. The common approach in these cases is to a) validate them under controlled conditions (i.e. laboratory samples) and b) conduct a sensitivity analysis and evaluate what can be compared against obtainable values in the "real world" setting. We follow this approach here. The robustness at this stage would refer to the prediction of FG abundance rather than any specific model itself. For instance, carbonyl predictions have Pearson correlation coefficients greater than

0.9, which we consider to be robust for trend estimation. As with complex models, more research is needed to improve and select a model that is consistent with available external observations.

- Section 3.6: This entire section, while interesting, seems like it would be better suited to go into the Supporting Information.

  We feel this short section is worth including in the main text as it highlights a particular group of samples that have not been identified in previous studies of FTIR analysis of ambient $PM_{2.5}$ — they are highlighted as worth noting in future studies (for measurement technique evaluation). We have now explicitly stated that identifying anomalous samples — particularly those pointed out by Ruthenburg et al. (2014) — as an objective of this paper.

  Technical Comments:

  We thank the reviewer for pointing these errors, which we have now corrected.

- P1L2: I'm not sure that FTIR needs to be defined, as it is commonly known acronym.

  We hope it becomes more commonly-known and used, but by journal conventions we are recommended to introduce acronyms when they are first introduced.

- P1L7: The phrase "including their model parameters" is parenthetical and could be enclosed by commas.

  Corrected.

- P7L9: OC is used here, but has not been previously defined.

  We have now defined the acronym OC in the abstract (page 1, lines 9: "organic carbon, OC, by..."), and then defined it again the first time it are used in the body of the paper (sentence: "or aggregated metrics such as OM or organic carbon (OC)").

- P1L11: It would be much more constructive to explain what the "series of possible input parameters" refers to. As it reads now, it is not instructive and, therefore, is unnecessary.

  We have deleted the sentence as we emphasize the same concept with the sentence: "It is possible to adjust model parameters (absorption coefficients for PF and number of latent variables for PLS) within limits consistent with calibration data to reduce these biases, but this analysis reveals that further progress in parameter selection is required."

- P1L14: TOR OC is used here, but it has not been defined.

  We have now defined the acronym TOR in the abstract (page 1, lines 9–10: "organic carbon, OC, measured by thermal optical reflectance method, TOR"), and then defined it again the first time it are used in the body of the paper (sentence: "network and estimates of OC agreed with collocated thermal optical reflectance (TOR) estimates within 90%.")

- P2L9: OM is used here, but it has not been previously defined.

  We have now defined the acronym in the same sentence: "organic mass (OM)".

- P2L27: Like the FTIR calibration, readers should be generally familiar with the most general principle of FTIR. This could be cut for brevity.

  We hope it becomes more commonly-known and used, but by journal conventions we are recommended to introduce acronyms when they are first introduced in both abstract and text.

- P3L23: Should FGss be FGs?

  Corrected.

- P4L16: PLS and PF "have," not "has"

  Corrected.

- P7L6: There should be a space before "A single parameter ..."

  Corrected.

- P10L16: The phrase "is depends" should be just "depends"

Corrected.

- P14L24: Should the phrase read "no gross overestimate, or that underestimation is unlikely?"

  We have rephrased the sentence to read: "Based on the assessment with PFr using a better-characterized absorption coefficient, comparisons with the reference ammonium values suggests that neither gross overestimation or underestimation of the fitting is likely."

- p19L5: There is an extra "a" at the beginning of this line.

  Corrected.

- p19L30: There should be a space after "Phoenix," and before "is."

  Corrected.

- Figure 10. The models do not need to be defined here as they are defined earlier and used often since.

  We have rephrased the caption from

  "PFo refers to peak-fitting using the original parameters. PLSr refers to partial least square using raw spectra. PFr refers to peak-fitting using the recalibrated absorption coefficients. PLSbc refers to partial least square using baseline corrected spectra. PLSbc* refers to partial least square using baseline corrected spectra is the same as PLSbc except that the number of LVs for aCH has been selected heuristically."

  to

  "PLSr and PLSbc refer to partial least square using raw and baseline corrected spectra respectively. PFo and PFr refers to peak-fitting using the original the recalibrated absorption coefficients. PLSbc* is the same as PLSbc except that the number of LVs for aCH has been selected heuristically."

- Figure 2. It might be clearer to use different colors in the bottom row to highlight the fact that you've used the test samples to validate the calibration made with the calibration samples.

  We think that using 12 more colors for differentiate between calibration and test samples will make the Figure difficoult to read. Therefore, we opted to add a label to each row to highlight that the first and second rows correspond to calibration and test.

- Figure 7. The urban x-axis value on the right side is overlapping with the rural x-axis on the left side. Also, is the correlation coefficient shown here R or $R^2$?

  Corrected. Moreover, the correlation coefficient is the Pearson correlation coefficient. We have changed the label of the Figure. "Pearson correlation coefficient".

- Figure 10. This figure would be much clearer if you centered the descriptions at the top (e.g., "From FG calibrations") and made them larger or bold.

  Corrected.

**Reviewer 2**

General comments

This manuscript details the similarities and differences between peak fitting and partial least squares as means of interpreting the organic functional group composition of ambient aerosol. The work is appropriate for AMT and relevant to the scientific community as filter samples are a routine way to collect atmospheric particles. Further FTIR is a non-destructive technique, so this method can be paired with other methods of analyzing filter samples. This work builds upon recently published advances in the area of FTIR spectroscopy. The paper presents the quantitative comparison of two methods and provides sufficient detail for readers to understand the merits of partial least squares as an approach to FTIR analyses. All assumptions are clearly explained. In many cases, sensitivity type analyses are offered as well. The abstract provides a good summary of the article itself. The figures are of high quality and the table and figures are easy to read and understand. With the incorporation of minor suggested changes below, I support this manuscript for publication.

We thank the reviewer for the encouraging feedback.

Specific comments

- The title is satisfactory as written but does not indicate that the paper is primarily about the method of functional group quantification and not about the results themselves.

  We thank the reviewer to pointing out that the title can be misleading. We have changed the title to

  "Analysis of functional groups in atmospheric aerosols by infrared spectroscopy: systematic intercomparison of calibration methods for US measurement network samples"

- The manuscript is somewhat difficult to read, likely owing to the use of many acronyms and exhaustive detail of methods and results. In many cases, the authors could improve the readability of the manuscript by occasionally using the unabbreviated term (PLS, LVs, PF, etc) before returning to abbreviations.

  We have added a table with abbreviations to improve the readability (Table A1 in Appendix A).

Technical corrections

  We thank the reviewer for pointing these errors, which we have now corrected.

- Pg 1 line 14: TOR used without definition

  Corrected.

- Pg 2 line 5: use "compose" instead of "comprise"

  Corrected.

- Pg 2 line 7: "methods . . .. include"

  Corrected.

- Pg 2, line 26: insert "and" before "ion chromatography"

  Corrected.

- Pg 11 line 15: I'm not sure if it should be "correspond" or "corresponds" please check the sentence meaning

  Corrected. ("correspond" because refers to "contributions to naCO")

  Done.

- Pg 11 line 31: I'm not sure what "evaluations of estimated quantities of using these absorption coefficients" means

  We rephrased this sentence as follows: we evaluate and discuss the FG estimated using the PF (using original and recalibrated absorption coefficients) and PLS (using baseline corrected and raw spectra) methods.

- Pg 13 line 23: define or redefine PFo and PFr

  (for PF) rephrased as: (for PF – PFo and PFr refer to PF using the original and the recalibrated absorbance coefficients respectively).

- Pg 13 line 26: define or redefine PLSr and PLSbc

  (for PLS) rephrased as: (for PLS – PLSbc and PLSr refer to PLS using the baseline corrected and unprocessed spectra respectively).

- Pg 14 line 12: missing word "are thought to be an upper bound"

  We have rephrased the sentence from

  "While our estimated reference values are thought to be upper bound on account of our assumptions — no evaporation loss of ammonium nitrate from PTFE or nitrate association with dust

instead of ammonium are not considered — the reference ammonium is overpredicted by the PSLr model at urban sites and PFo for both urban and rural sites."

to

"While our estimated reference values are thought to be the upper bound of ammonium concentrations (on account of our assumptions that (i) there are no evaporation loss of ammonium nitrate from PTFE nor (ii) nitrate association with dust instead of ammonium), the reference ammonium is overpredicted by the PSLr model at urban sites and by the PFo model at both urban and rural sites."

- Pg 14 lines 13-14: awkward sentence "no evaporation loss of ammonium nitrate from PTFE or nitrate association with dust instead of ammonium are not considered"

  This point is now addressed in previous comment.

- Pg 14 line 25: do the authors mean "is likely"? Otherwise it is a double-negative

  We have rephrased the sentence from

  "Based on the assessment with PFr using a better-characterized absorption coefficient, comparisons with the reference ammonium values suggests that no gross overestimation or underestimation of the fitting is unlikely."

  to

  "Based on the assessment with PFr using a better-characterized absorption coefficient, comparisons with the reference ammonium values suggests that neither gross overestimation or underestimation of the fitting is likely."

- Pg 15 line 20: use "compose" instead of "comprise"

  Corrected.

- Pg 15 line 22: "vary" not "varies"

  Corrected.

- Pg 15 line 23: "correlation . . . is"

  Corrected.

- Pg 15 line 25: coefficient should be plural as it refers to two (14.84 and 8.89)

  Corrected.

- Pg 15 line 26: "correlation . . . is"

  Corrected.

- Pg 16 line 4: what is areal density?

  We have rephrased this sentence from

  "Figure 8 compares the number of areal density of tCO and cCOH predicted by the same methods."

  to

  "Figure 8 compares the abundance of tCO and cCOH predicted by the four methods."

- Pg 16 line 27: I think the authors mean "such low abundances"

  Corrected.

- Pg 19 line 5: omit "a" before "another"

  Corrected.

- Pg 19 line 30: insert a space between "Phoenix, is"

  Corrected.

---

## Author Comment (AC2) · 9 Mar 2019

The comment was uploaded in the form of a supplement:
https://www.atmos-meas-tech-discuss.net/amt-2018-331/amt-2018-331-AC2-supplement.pdf